# Gap Junction Channel Regulation: A Tale of Two Gates—Voltage Sensitivity of the Chemical Gate and Chemical Sensitivity of the Fast Voltage Gate

**DOI:** 10.3390/ijms25020982

**Published:** 2024-01-12

**Authors:** Camillo Peracchia

**Affiliations:** Department of Pharmacology and Physiology, School of Medicine and Dentistry, University Rochester, Rochester, NY 14642-8711, USA; camillo.peracchia@gmail.com or camillo_peracchia@urmc.rochester.edu

**Keywords:** gap junctions, channels, chemical gating, voltage gating, calmodulin, connexins, innexins

## Abstract

Gap junction channels are regulated by gates sensitive to cytosolic acidification and trans-junctional voltage (Vj). We propose that the chemical gate is a calmodulin (CaM) lobe. The fast-Vj gate is made primarily by the connexin’s NH_2_-terminus domain (NT). The chemical gate closes the channel slowly and completely, while the fast-Vj gate closes the channel rapidly but incompletely. The chemical gate closes with increased cytosolic calcium concentration [Ca^2+^]_i_ and with Vj gradients at Vj’s negative side. In contrast, the fast-Vj gate closes at the positive or negative side of Vj depending on the connexin (Cx) type. Cxs with positively charged NT close at Vj’s negative side, while those with negatively charged NT close at Vj’s positive side. Cytosolic acidification alters in opposite ways the sensitivity of the fast-Vj gate: it increases the Vj sensitivity of negative gaters and decreases that of positive gaters. While the fast-Vj gate closes and opens instantaneously, the chemical gate often shows fluctuations, likely to reflect the shifting of the gate (CaM’s N-lobe) in and out of the channel’s pore.

## 1. Introduction

Evidence that “direct” cell-to-cell communication can be decreased down to complete cell-to-cell uncoupling was discovered accidentally nearly a century before gap junction mediated cell–cell communication was discovered. Indeed, almost one and a half century ago T.W. Engelmann discovered that wounded heart cells become independent from adjacent cells as they die [1]. Engelmann’s discovery was based on evidence that when the heart fibers are damaged the injury-potential quickly disappears. This phenomenon, named “healing-over”, revealed a major difference between cardiac and skeletal muscle fibers—in Engelmann’s writing: “… herzmuskelzellen leben zusammen und sterben einzeln (cardiomyocytes live together and die alone)”. This finding startled him as the cardiac muscle was thought to be a cellular syncytium—cell–cell communication via protoplasmic bridges [2]. The syncytium hypothesis was eventually disproven by electron microscopic studies. Engelmann’s healing-over phenomenon was finally confirmed almost three quarters of a century later [3,4,5,6]. Healing-over (cell–cell uncoupling) is a property of all organs made of coupled cells and is caused by the gating mechanism of gap junctions; rev. in [7,8,9,10].

Almost a century after Engelmann’s study, J. Délèze discovered that damaged cardiac fibers do not heal in the absence of extracellular calcium (Ca^2+^_o_), but do so with Ca^2+^-addition to the bathing medium [11], as Ca^2+^ enters the broken cell and rises the cytosolic Ca^2+^ concentration [Ca^2+^]_i_ [12,13]. This pivotal observation [11], published a few years after most cells were found to communicate electrically and metabolically with neighboring cells [14,15], indicated that Ca^2+^_i_ is a key player in gap junctions’ chemical gating. The Ca^2+^_i_ role in cell-to-cell uncoupling was confirmed by data demonstrating that uncoupling parallels an increase in [Ca^2+^]_i_ at cell–cell junctions; the [Ca^2+^]_i_ rise was proven by intracellular injection of the Ca^2+^-sensitive indicator aequorin [16,17].

In the early 1970s, Ca^2+^_i_ was thought to be the only regulator of the permeability of gap junction channels. This, however, was questioned in 1977 by evidence that cytosolic acidification induced by treatment with solutions gassed with 100% CO_2_ uncouples embryonic cells of *Xenopus laevis* [18,19]. This finding seemed to suggest that hydrogen ions (H^+^_i_) play a key role in channel gating [20], but in the following years this hypothesis has been seriously questioned [21,22]. However, evidence for a direct role of H^+^ in gating has been reported in connexin46 (Cx46) hemichannels [23].

In the early 1980s we reported that uncoupling by cytosolic acidification is inhibited by treatment with calmodulin (CaM) blockers [24], suggesting that gating by acidification is likely to be mediated by [Ca^2+^]_i_ above normal values, and consequential activation of CaM; our data were further confirmed by early data reporting the presence of CaM sites in gap junction proteins [25,26]. In agreement with a role of Ca-CaM in cell–cell uncoupling is also evidence that low [Ca^2+^]_i_ are effective in gating gap junction channels [12,13,17,22,27,28,29,30,31,32,33,34,35,36,37,38,39].

Based on numerous subsequent studies that confirmed in different cell systems the CaM-gating hypothesis [38,40,41,42,43,44,45,46,47,48,49,50,51,52], we believe that cell–cell uncoupling is mediated by Ca^2+^-activated CaM. Indeed, in addition to functional and biochemical studies cited in the previous, in a study on connexin36 (Cx36) the CaM binding to the COOH-terminus of Cx36 was proven by SOLUTION NMR (PDB-2n6a; L.W. Donaldson) [53]. The CaM-Cx45 binding was also proven in living cells by Bioluminescence-Resonance-Energy-Transfer (BRET) [49]—the binding was [Ca^2+^]_i_-dependent and was blocked the CaM inhibitor W7. High resolution NMR spectroscopy demonstrated that the CaM interaction with the CaM binding site of Cx45 causes global structural changes in (15)N-labeled CaM without altering the size of the structure [49]. CaM was also proven to directly bind in a Ca-dependent way to Cx32′s peptides matching the sequences of CaM sites at amino-terminus (NT) and CT domains [43]. In addition, CaM binding to Cx32 was reported to partially prevent Cx32′s phosphorylation caused by the EGF-activation of tyrosine kinase in solubilized membranes of rat liver [54].

Recently, peptides matching the CaM-binding sequence of the second half of the cytoplasmic loop (CL2) of connexins like Cx32, Cx35, Cx43, Cx45 and Cx57 were found to bind CaM in a Ca-dependent and independent manner [44]. These data confirmed an earlier report that tested a synthetic peptide sequence matching the CL2′s CaM-binding sequence of Cx43 (res. 144–158) by using an X-ray scattering technique of Ca^2+^-induced changes in conformation [55]; with peptide binding, CaM assumed a more globular structure, suggesting that CaM binds to the peptide in a “collapsed” state [55]. While we feel that there is strong evidence for a Ca-CaM role in chemical gating, it should be stressed that the CaM-based gating mechanism is yet to be unequivocally proven. In addition to Ca-CaM, there is evidence that the channels can be closed by lipophilic volatile anesthetics and long chain alcohols [56,57,58,59]. Indeed, the use of anesthetics enabled the earliest single channel recordings for most connexins [60,61,62].

## 2. Gap Junction Channel Gates

The channels of gap junctions possess two gates that are sensitive to changes in the ionic composition of the cytosol and trans-junctional voltage (Vj); rev. in [10,63,64,65,66,67]. Our hypothesis is that the chemical gate is a CaM lobe—probably the N-lobe, whereas the fast-Vj gate is believed to be made mostly of the NH_2_-terminus domain (NT) of gap junction connexins/innexins [64,68].

In all connexins/innexins channels the chemical gate closes hemichannels at the negative side of Vj [69], which may be consistent with the fact that the CaM lobes have a net negative charge [70] and the channel’s pore (vestibule) is positively charged. In contrast, the gating polarity of the fast-Vj gate is connexin dependent [64] and is related to NT’s charge type. In the connexin/innexin channel with a positively charged NT domain (Cx32, Cx38, and Cx43, for example) the hemichannel closes at the negative Vj side (negative gaters), while in those with a negatively charged NT domain (Cx26, Cx40, Cx50, and Cx37, for example) the hemichannels close at Vj’s positive side (positive gaters) [64,71].

The connexins listed below are primarily expressed in the following tissues:Cx32: hepatocytes, exocrine pancreas, kidney, myelinated Schwann cells.Cx38: *Xenopus* cells.Cx43: heart ventricular and atrial cardiomyocytes, endothelial cells, smooth muscle, and fibroblasts.Cx26: liver.Cx40: cardiomyocytes of atria and cardiac conduction system (His bundle and upper bundle branches).Cx50: lens epithelium, retina (horizontal cells and Müller cells), spinal cord tissue.Cx37: brain, uterus, ovary, endothelial cells of blood vessels.For thorough review articles on connexin expression in vertebrate organs see [46,64,71,72,73,74,75,76].

The electrical behavior of connexins’ gates is demonstrated in individual channels studied by dual-whole-cell-clamp (DWCC) if cell coupling very low, such as in minimally coupled cells, during new channel formation, or at the latest stages of cell–cell uncoupling (Figure 1). In 1997, Bukauskas and I reported the gating behavior of single channels of fibroblasts and HeLa cells transfected with Cx43 during exposure to solutions bubbled with 100% CO_2_ (Figure 1) [77]. Junctional current (Ij), conductance of single channels (γj) and kinetics of Ij were recorded during uncoupling and recoupling at different Vj’s, which allowed us to distinguish the activity of the chemical gate from that of the fast-Vj gate. Since in Cx43 channels the fast-Vj gate is activated only by Vj’s > 40–50 mV, by studying channel gating at different Vj values, such as Vj = 30 mV (fast-Vj gates opened), Vj = 55 mV (fast-Vj gates flickering) and Vj = 70 mV (fast-Vj gates mostly closed), the activity of each of the two gates could be monitored in detail [77].

At a Vj of 55 mV, Vj induces fast Ij flickering (Figure 1A) from the open state, γj(main state), to the residual state, γj(residual), with a ratio γj(residual)/γj(main state) of 20–25% (Figure 1; activity of the fast-Vj gate), as well as slow transitions between open and totally closed states (Figure 1; activity of the chemical gate) [77]. This indicates that chemical gates and fast-Vj gates operate independently. At the end of the CO_2_ treatment, individual channels reopen by a slow transition from the closed to open state (Figure 1A(b); calmodulin gate reopening), followed by fast Ij-flickering from the open to residual state (Figure 1A; activity of the fast-Vj gate). Note that the size of the residual state has been reported to vary slightly among connexins [78,79].

Significantly, the slow transitions of the chemical gate frequently show fluctuations, during both channel closing and opening events (Figure 1B(a,b), red arrows). The fluctuations may agree with our “cork-gating” hypothesis, as they suggest that a particle, a CaM lobe? (Figure 1B(c)), moves momentarily in and out of the channel’s pore before assuming the final state: fully closed or fully open channel.

These data [77] are consistent with evidence that gap junction channels have two major gating mechanisms and may confirm our hypothesis that CO_2_ causes channel gating by activating a gate made of a CaM-lobe. Our single channel data findings [77] confirmed earlier data obtained in insect cells [80] and in mammalian cells expressing Cx40 [81]. For detailed descriptions of single-channel behavior, kinetics, and models of channels’ fast-Vj gating, see ref. [78,79,82,83,84,85].

## 3. Cork Model of Chemical Gating

For explaining our view on the behavior of the chemical-gate we will first describe the gating model that we have proposed several years ago [52]. The model, named “Cork Gating” (Figure 2), proposes that channel gating is caused by the plugging of the channel’s pore by a CaM lobe, as a “cork” would plug a bottle.

This model considers two gating mechanisms. One, named “Ca-CaM-Cork”, suggests that gating is initiated by Ca^2+^-induced activation of the CaM’s N-lobe; note that cytosolic acidification increases [Ca^2+^]_i_; rev. [45]. The other, “CaM-Cork”, suggests that channel gating takes place without [Ca^2+^]_i_ rising above normal values and is activated by significantly large Vj values negative at the site of the gating hemichannel. Ca-CaM-Cork gating is reversed by a recovery of resting [Ca^2+^]_i_ values, whereas CaM-Cork gating is reversed by a return to Vj = 0 or by Vj positive at the gated hemichannel. CaM is believed to be linked to connexins/innexins by its C-lobe in Ca^2+^-independent way [44]. With a rise in [Ca^2+^]_i_ we think that the N-lobe gates the channel by plugging the channel’s pore, and probably affecting connexin conformation as well (Figure 2C).

### 3.1. Ca-CaM-Cork Gating Model

This mechanism proposes that a rise in [Ca^2+^]_i_ over resting level activates the N-lobe of CaM, allowing it to interact with a connexin site and plug the channel’s pore (Figure 2C). At normal cytosolic [Ca^2+^]_i_, CaM is thought to be bound to connexins by the C-lobe to the CL2 site, while the N-lobe is believed to be unbound (Figure 2C). The Ca^2+^-affinity of the C-lobe is stronger than that of the N-lobe by nearly one order of magnitude [86,87]. Therefore, the N-lobe is thought to interact with the connexin’s gating site, CL2 or NT-terminus (NT), with a rise in [Ca^2+^]_i_ over resting values, plugging the channel’s pore and probably changing connexins’ conformation as well.

### 3.2. CaM-Cork Gating Model

The potential CaM-Cork gating mechanism was manifested at first by the activity of heterotypic mutant-Cx32 channels [69]. We propose that some connexin mutations may able to allow the negatively charged N-lobe of CaM to access the channel’s pore and plug it by binding electrostatically with the positively charged pore even at resting [Ca^2+^]_i_ or [H^+^]_i_. The N-lobe would be displaced from the pore by Vj positive at the mutant side, causing channel opening. Significantly, large Vj gradients close the channels by the chemical gate at the Vj negative side also in homotypic channels made by wild type connexins in the absence of uncouplers [88] (see below). It is worth noting that unapposed hemichannels are opened by membrane depolarization in all connexins tested regardless of whether they are positive or negative fast-Vj gaters [89,90]. This may be consistent with the role of a CaM-cork gate which would open and close at Vj positive and negative, respectively, in all connexin channels. For a detailed description of the gating mechanism of unapposed hemichannels see [91,92,93,94,95,96].

Both the positively charged channel’s pore (Figure 2A,C) and the negatively charged CaM lobes (Figure 2B) are ~25 × 35 Å in diameter. Thus, a CaM lobe would be expected to be able to bind and fit easily in the positively charged connexon’s pore (Figure 2C).

## 4. The Chemical Gate Is Voltage Sensitive

In 1996, while we were studying by using double voltage-clamp electrophysiology of the gating of connexin32 (Cx32) channels expressed in *Xenopus* oocyte pairs, we found that CL2 contains a domain most important in determining CO_2_ gating sensitivity [97]. In contrast, 84% of the COOH-terminus domains did not seem to play a significant gating role [98], although positively charged residues of CT (CT1) appeared to decrease CO_2_ gating sensitivity [98]. The absence of a role of most of the CT in chemical gating of the Cx32 channel confirmed an earlier report by Werner and coworkers [99]. [99]. Incidentally, also in Cx43 channels the truncation of most of the CT domain in mutant Cx43M257 (Cx43K258stop) does not affect the chemical gating mechanism [28].

To probe in more detail Cx32′s channel gating, we constructed three mutants [69]. One mutant, tandem, is a dimer made of two Cx32 monomers linking NT-to-CT (Figure 3B, inset). Another mutant, 5R/E, is a Cx32 molecule in which 5 CT1 positively charged residues, R215, R219, R220, R223, and R224, are replaced with a negatively charged residue (E). In the third mutant, 5R/N, the five positively charged CT1 residues are replaced with a neutral residue (N). Tandem and 5R/E mutants do not express functional channels when they are paired homotypically in *Xenopus* oocytes, but in heterotypic pairs tandem-Cx32 or 5R/E-Cx32 (tandem-32 and 5R/E-32, respectively) minimal coupling develops. In contrast, 5R/N mutants are functional homotypically (5R/N-5R/N) and when they are heterotypically paired with the Cx32 wild-type (5R/N-32).

These mutants, heterotypically paired with Cx32 in *Xenopus* oocytes, provided us with evidence that the gap junction’s chemical gate is also Vj-sensitive, and revealed the existence of a ‘‘slow Vj-sensitive chemical gate’’ distinct from the conventional fast-Vj gate [69]; note that the slow Vj-sensitive chemical gate has also been named “loop gate” [100]. Since the heterotypic channels tandem-32 and 5R/E-32 behaved similarly, we are focusing here primarily on the behavior of tandem-32 channels.

### 4.1. Vj-Ij Behavior of Heterotypic Tandem-32 Channels

Homotypic Cx32 junctions (32–32) display a typical Vj sensitivity, as Ij exponentially decreases for *V*j > ±40 mV (Figure 3A,B) [69], while tandem-32 channels display an unusual Ij-Vj behavior (Figure 3A,B) [69]. With tandem at the negative side of Vj, as Vj is raised in 20 mV steps from 20 to 120 mV, both initial and final Ij progressively decrease to very low values, and the channels are sensitive to Vj even at the lowest Vj values (Figure 3A), as the chemical gates close. In contrast, with Vj positive at the tandem side Ij gradually increases to high values, and the Ij at the end of the pulse is bigger than the Ij initial up to Vj = 80 mV (Figure 3A), as tandem’s chemical gates open. The asymmetrical Ij-Vj behavior of tandem-32 and other mutant-32 channels is clearly shown in plots of normalized Gj: Gj-final/Gj-initial (Gj-fin/Gj-ini) versus Vj (Figure 3B). However, note that at Vj = 100–120 mV, tandem-side positive, the behavior is more complex as it is multiphasic (Figure 3A). The reason for it is that while chemical gates open at tandem side, some fast-Vj gates and chemical gates close at Cx32 side. Indeed, high Vj gradients activate the chemical gates also in Cx32 wild type channels [88] (see below). This unusual Ij-Vj behavior indicates that Vj positive or negative at the tandem side gradually opens or closes, respectively, a greater number of channels by a slow gate, and we believe it is the Vj-sensitive chemical-gate (CaM’s N-lobe? Figure 2C).

### 4.2. Behavior of Channels Made of Tandem-32 Subjected to 60 mV Vj Pulses

To further test the Vj activity of tandem-32 channels, 60 mV Vj pulses (positive at tandem side) of 20 s duration were repeatedly applied at 45 s intervals [69]. Three different Ij behaviors were manifested (Figure 4): 1. Rapid Ij rise followed by slow Ij rise (Figure 4, #1–3 pulses) 2. Triphasic Ij time-course (Figure 4, #4–9 pulses); 3. Conventional Ij behavior (Figure 4, pulses #10–18). The testing of the normal Vj protocols (positive at tandem side) following the application of repeated *V*j pulses resulted in Ij-Vj activity like that of 32–32 channels (Figure 4, pulses 19–27); however, note that pulse #21 (Vj = 60 mV) generates an Ij time course that is biphasic, as that of pulse #6, indicating that, following the 60 mV Vj pulses, the chemical gates of some of the channels have closed.

In Figure 4, the red arrows at pulses #1–8 point to the Ij transition from rapid to slow Ij rise. The rapid Ij rise reflects the Ij of channels that were already open before the pulse, while the slow Ij rise reflects the Vj-induced opening of previously closed channels. This description has focused on the behavior of the chemical gate, but what about that of the fast-Vj gate? Indeed, a few fast-Vj gates are also activated, as Vj gradients of 60 mV activate the fast-Vj-gate of 30–40% of the channels (Figure 3B). The fast-Vj-gate of Cx32 channels is known to be sensitive to Vj negative, as Cx32 channels are a negative gaters [101], therefore 60 mV Vj pulses positive at the tandem side also activate the fast-Vj gate of a 30–40% of Cx32’s hemichannels. Indeed, the behavior of the fast-Vj gate manifests itself starting at pulse #4, which shows the first triphasic Ij time course (Figure 4), while the behavior of the fast-Vj-gate at pulse 1–3 is masked by the overwhelming Ij rise due to the opening of chemical gates. Indeed, when all the channels have become operational, starting from pulse #9 onward, only the behavior of Cx32’s fast-Vj gates is manifested, represented by an Ij that exponentially decays to a steady state level (Figure 4, blue arrows).

### 4.3. Behavior of Channel Made of Tandem-32 Subjected to Steady State Vj

Data showing that the application long Vj-pulses of opposite polarity gradually increases or decreases Gj in channels made of tandem-32 (Figure 3) [69] indicated that these heterotypic channels may be sensitive to steady-state Vj as well. In this case, the application of steady-state Vj may be instrumental in revealing the magnitude and kinetics of this phenomenon.

In heterotypic channels made of tandem-32 and initially voltage-clamped at a Vm = −20 mV (Vj = 0 mV), the application of a steady-state Vj of 40 mV (positive at tandem side) exponentially increases Gj by 262 ± 64% (mean 6 SD, n = 4), with a time constant (τ) of 0.88 ± 0.2 min (n = 4)(Figure 5A,B); Gj was monitored by applying pulses of 20 mV and 2 s duration to one of the two oocytes at 10 s intervals [69]. Vj gradients of 10 and 20 mV (tandem side positive) increase Gj by 10–20% and 80–130%, respectively [69]. With a reversal of Vj to 40 mV (negative at tandem side, Figure 5B), Gj exponentially drops to a level lower than control values (Figure 5B), suggesting that the channels close at the negative tandem side.

Significantly, upon returning to Vj = 0 mV from 40 mV Vj (positive at tandem side), Gj suddenly rises before decreasing (Figure 5A). This behavior is likely to reflect the fast reopening of few chemical and fast-Vj gates of the Cx32 hemichannels (note that the chemical gates of Cx32 are also Vj sensitive [88], albeit much less so than those of the tandem; see below). Simultaneously, chemical gates start closing at the tandem side (Figure 5A). Indeed, the sudden Gj rise is not seen with Vj reversal from the tandem side positive to negative (Figure 5B) because while the gates of one hemichannel open they close at the opposite hemichannel.

The reason the chemical gates are closed at the tandem hemichannels without application of chemical uncouplers is not clear. Perhaps, while in wild-type Cx32 hemichannels both the six NT and CT domains are free, in tandems three of them are linked to the neighboring tandem (Figure 3B, inset). Our working hypothesis is that in Cx32 hemichannels the six NT and CT free domains somehow protect the channel’s pore, preventing the entrance of the plugging molecule. The interlinkage of these domains in tandem hemichannels (Figure 3B, inset) may leave the pore of the channel unprotected and accessible to the gating molecule (CaM-lobe?).

### 4.4. The Chemical-Gate of Homotypic Cx32 Channels Is Also Sensitive to Vj

On the basis of the data presented above, in the early 2000s we thought that the chemical gate might not be active in the absence of uncoupling treatments or connexin mutations. In 2007, however, we found that the chemical gate of homotypic Cx32 channels and homotypic channels made of a Cx32 mutant in which the COOH-terminus was delated at residue #225 (Cx32-D225), can be turned on by large Vj gradients and forced to close the channel, most likely by plugging the channel’s pore (cork-gating), without application of uncoupling treatments (Figure 6 and Figure 7) [88].

In oocytes expressing homotypically Cx32 channels, the application of Vj pulses of –100 mV (12 s duration, repeated at 30 s intervals), causes the Gj peak (GjPK) and Gj steady-state (GjSS) to gradually decrease by 50–60%, with a τ of ~1.2 min (Figure 6A,C–E). GjPK decreases more drastically (Figure 6C), as GjSS/GjPK rises from ~0.4 to ~0.6 (Figure 6D). Milder effects are seen with pulses of –60 mV Vj [88]. Gj, measured during recovery by testing small pulses of Vj (20 mV, 2 s duration, repeated at 30 s intervals), gradually recovers, usually achieving greater than initial values, following an exponential rise with a τ of ~7 min (Figure 6E).

When the train of Vj pulses (−100 mV) passed to oocyte #1 are instantly followed by a similar train of Vj pulses (−100 mV) passed to oocyte #2 (Vj polarity reversal) (Figure 7A), both GjPK (Figure 7B) and GjSS slightly increase briefly, as compared to the last pulses of the previous train, before dropping. Since fast-Vj gates and chemical gates of Cx32 both close at Vj negative [64,102], by reversing the Vj polarity the closed hemichannel gates, now exposed to Vj positive, start opening, whereas the gates at Vj negative start closing [88]. Obviously, if the kinetics of opening and closing were identical, no change in GjPK and GjSS would be expected from the last pulse of the earlier train to the first pulse of the new train. However, since GjPK and GjSS are greater after the Vj polarity reversal it is likely that the positive Vj has a greater and faster effect at opening hemichannels than negative Vj does at closing them.

GjSS/GjPK drops significantly when negative Vj pulses are switched to oocyte #2 (Figure 7C) [88]; a likely reason for the apparent rise in Vj sensitivity is that the hemichannels in oocyte #2 still possess a normal Vj sensitivity, as they have not been subjected to repeated Vj-negative pulses (e.g., 60% of the hemichannels are Vj-sensitive, as in the example given above), while most of the oocyte #1 hemichannels are part of the remaining partially Vj-insensitive channels. Even more drastic results are seen with the Cx32 mutant Cx32-D225 (Figure 6B–E and Figure 7B,C) [88], suggesting that over 80% of CT may not be involved in Vj gating.

Data showing that the chemical gate is active in homotypic Cx32 channels proves that this gate, probably the CaM’s N-lobe, can be induced to be active even in the absence of uncoupling treatments or Cx mutations. Our data [88] confirm an earlier study on the activity of single Cx32 channels which reported that sporadic slow-gating events, slow channel closure to zero conductance state, typical of the chemical gate, are witnessed in cells expressing Cx32 channels subjected to Vj pulses of −60 mV [103]. In our report, −60 mV Vj pulses were also found to activate the chemical-gate [88]. With -60 mV Vj pulses, GjPK drops slightly in channels made of Cx32 or Cx32-D225. GjPK drops to 82.1± 3.7% (mean ± SE, n = 4) and 78.8 ± 7.0% (mean ± SE, n = 6) in Cx32 and Cx32-D225, respectively, following single exponential decays with τ = 2.6 and 2.9 min, respectively. GjSS minimally drops, such that the GjSS/GjPK ratio grows from 0.69 ± 0.02 to 0.71 ± 0.006 (mean ± SE, n = 4) in channels made of Cx32 and from 0.63 ± 0.07 to 0.67 ± 0.05 (mean ± SE, n = 6) in channels made of Cx32-D225—increases of 3% and 6%, respectively [88].

Significantly, deletion of most of the CT in the Cx32D225 mutant renders the channels more sensitive to Vj (Figure 6 and Figure 7). A possible reason for it is that while in wild-type Cx32 hemichannels the CT domain may partially protect the pore of the channel, preventing the access of the plugging CaM-lobe, its absence in the Cx32D225 mutant may leave the pore of the channel less protected and so more easily accessible to the plugging molecule (CaM’s N-lobe?), as in the case of the tandem mutant (see above).

### 4.5. Effect of Vj Steady-State on Gj during Uncoupling by CO_2_

Heterotypic tandem-32 (Figure 8A) or 5R/E-32 (Figure 8B) channels were exposed to Vj = 40 mV of different polarity during superfusion with salines gassed with 100% CO_2_ [69]. In both tandem-32 and 5R/E-32 cell pairs, Gj, decreased to very low values with CO_2_ treatment at Vj = 0 mV, reversibly increased with Vj = 40 mV, positive at mutant-side (Figure 8A,B). In contrast, Vj negative the mutant side drastically and reversibly reduced Gj to nearly zero (Figure 8A,B). Significantly, in mutant-32 channels positive Vj at the mutant side was progressively less effective in raising Gj as uncoupling developed (Figure 8A,B) and progressively more effective during recovery (Figure 8A,B). This is most obvious in 5R/E-32 channels in which Vj is minimally effective in opening channels at maximum uncoupling (Figure 8B). We propose that the most obvious reason for this phenomenon is that there are two populations of channels: one in the Ca-CaM-cork gating state (Figure 8C, top) and the other in the CaM-cork gating state (Figure 8C, bottom). In the CaM-cork gated state the tandem hemichannel gates are opened by Vj positive (Figure 8C, bottom), while those in the Ca-CaM-cork gated state are not (Figure 8C, top). On the basis of our hypothesis that CaM’s N-lobe is the gating structure, we think that in the CaM-cork gating state the CaM lobe (N-lobe) may close the mutant hemichannel by binding only electrostatically with the positively charged connexon’s pore, while in the Ca-CaM-cork gating state the Ca^2+^-activated CaM lobe may interact with the connexin’s CaM site electrostatically and hydrophobically. Therefore, while the CaM-cork state is likely to be reversed by Vj positive, the Ca-CaM-cork is not reversed by Vj, as it can only be reversed by the reestablishment of resting pH_i_ values (indeed, resting [Ca^2+^]_i_). In other terms, as uncoupling progresses, we believe that a greater number of mutant hemichannels are switched from the CaM-gated state to the Ca-CaM-gated state. In channels made of mutant-32, the chemical gating occurs mostly at the mutant hemichannel because all the Cx32 mutants tested were found to be much more sensitive to CO_2_-induced uncoupling than Cx32 channels. In fact, 3–5 min treatment with 100% CO_2_ reduces the Gj of 32–32 channels by only ~15% [69,104], while it decreases by ~100% the Gj of tandem-32 and 5R/E-32 channels (Figure 8A,B) [69,104].

### 4.6. CaM-Expression Inhibition Affects the Vj-Sensitivity of the Chemical Gate of Tandem-32 Channels

For testing whether inhibiting CaM expression affects the Vj-sensitivity of the chemical gate, tandems were expressed in oocytes injected with oligonucleotides antisense to CaM, and tandem-expressing oocytes were paired heterotypically with oocytes expressing Cx32 (tandem-32 channels) [104]. CaM mRNA is significantly degraded 5 hr after oocyte injection with oligonucleotides antisense to the CaM mRNA [105]. The oocyte pairs were tested with the conventional Vj protocol (Vj steps of 20 mV, ±120 mV max., and 25 s duration, applied at 45 s intervals).

Figure 9 shows that the asymmetry of the Vj behavior of tandem-32 channels (Figure 9A) is abolished or mostly reduced following CaM-expression inhibition (Figure 9B). In fact, plots of normalized Gj, Gj-final/Gj-initial (Gj-fin/Gj-ini) versus Vj for 32–32 and tandem-32 (anti-CaM) channels are virtually superimposable (Figure 9B), were not for a very small asymmetry of tandem-32 with positive Vj at tandem side (Figure 9B).

### 4.7. CaM-Expression Inhibition Affects the Sensitiviti to Vj of the Chemical Gate in Cx45 Channels

Channels made of homotypic Cx45 are quite sensitive to Vj, such that their conductance is decreased by Vj gradients of ±5 mV or lower (Figure 10A,B) [106,107,108]. Different from most connexin channels, in Cx45 the gate affected by Vj is primarily the chemical gate; the fast-Vj gate is also expressed, but large Vj gradients are needed for its activation [109].

Data from heterotypic channels made of Cx45 and Cx43 linked to the Enhanced Green Fluorescence Protein (Cx45 paired to Cx43-EGFP), indicated that some channels are closed by the chemical gate also at Vj = 0 mV [109]. The slow-Vj-gating behavior of Cx45 channels was shown at the single-channel level by data demonstrating slow and complete channel closure with small Vj gradients [109,111]. This behavior, typical of the chemical gate [77], was also observed in Cx45 hemichannels [112]. The significant difference in Vj sensitivity between the chemical gate and fast-Vj gate of Cx45 channels renders this connexin ideal for studying the chemical gate behavior independently from that of the fast-V gate.

While with normal expression of CaM, Ij decays for Vj values larger than ±5 mV (Figure 10A,B), with CaM-expression inhibition, Ij does not decay with Vj values smaller than ±40 mV (Figure 10C,D), and the τ values at Vj = 100 mV and Vj = 120 mV, are ~14 s and ~11 s, respectively [110]. The vastly reduced Vj sensitivity with CaM-expression inhibition is demonstrated in plots of Gjss/Gjmax versus and Vj, where the Boltzmann values are V_0_ = 75.5 mV, η = 1, Gjmin = 0.49, and Gjmax = 1.03 (Figure 10D) [110].

CaM-expression inhibition also largely reduces the sensitivity to CO_2_ of Cx45 channels. A 15-min application of 100% CO_2_ causes Gj to drop reversibly by ~17%, at a maximal rate of ~3.3%/min, while it drops Gj of controls to nearly 0% (Figure 11) [110]. CaM-expression inhibition also reduces the CO_2_-induced gating of oocytes expressing homotypic Cx38 channels [113], as well as heterotypic mutant-32 channels [104]. The brief Gj rise that precedes the Gj drop in controls (Figure 11) is not seen with CaM-expression inhibition; the initial Gj rise may be caused by activation of Ca^2+^/CaM kinase II resulting from a small increase in [Ca^2+^]_i_, while the following greater [Ca^2+^]_i_ rise may close the channels by the Ca-CaM-Cork gating mechanism (Figure 11). The role of Ca^2+^/CaM kinase II in opening connexin channels has been reported in various cells [51,114].

Consistent with this interpretation are data on unapposed Cx32 hemichannels, which showed that a [Ca^2+^]_i_ rise to ~500 nM causes hemichannel opening [115,116,117]; this small rise in [Ca^2+^]_i_ induced by treatment with 2 mM A23187 (a Ca^2+^-ionophore), resulted in ATP release and dye uptake which was dependent on Cx32 expression and blocked by a Cx32 mimetic peptide [115]. The opening of hemichannels was prevented by treatment with 20 μM W7 (a CaM-blocker) [115], indicating that CaM may play a key role in opening hemichannels. A later study confirmed these data by testing a [Ca^2+^]_i_ rise and inhibition of CaM in unapposed hemichannels made of Cx43 [116]. Significantly, an increase in [Ca^2+^]_i_ in the 500 nM range opens unapposed hemichannels, but a greater [Ca^2+^]_i_ rise closes them completely, probably by activating the Ca-CaM-Cork gating mechanism.

### 4.8. CaM Mutants Affect Gating Sensitivity and Expression of Cx32 Channels

Further evidence that CaM is linked to gap junctions and plays a direct role in gating was provided by experiments testing a CaM mutant in which the N-terminal EF-hand pair was replaced by the C-terminal pair. Since the Ca^2+^ affinity of CaM’s C-terminal EF hand pair is greater by nearly one order of magnitude than that of the N-terminal pair [86], we thought that at the greater CaMCC ‘s Ca^2+^-binding could translate into greater increase in gap junctions’ chemical gating sensitivity.

Indeed, CaMCC was found to drastically enhance the gating sensitivity of gap junction channels made of homotypic Cx32 [52]. The greater gating sensitivity, most likely caused by the higher overall Ca^2+^-binding affinity of this mutant, was only observed when the mutant was expressed before Cx32, indicating that the mutant interacts with Cx32 before gap junction formation. Immunofluorescence data [52] further demonstrated that this interaction incorporates native or mutant CaM in the connexon as an integral subunit.

The increased gating sensitivity caused by the expression of CaMCC was further confirmed by testing the exposure to 100% CO_2_, which drops the pH_i_ to ~6.3 and causes a rise in [Ca^2+^]_i_. With CO_2_, Gj quickly drops to zero, while in controls it drops by only ~15%. During CO_2_-washout, Gj remains indefinitely at zero, but reversibly recovers by superfusing with 180 µM BAPTA, which decreases [Ca^2+^]_i_ to values lower than resting values [52].

We also tested the expression of CaM mutants missing one or more of the four high-affinity Ca^2+^-binding sites. In these CaM mutants, glutamates (E) most relevant for Ca^2+^-binding were replaced with alanines (A) in CaM’s EF-hand sites. These mutations dramatically reduce the Ca^2+^-affinity of the EF-hands [118]. The expression of CaM 1,2,3,4 (E32A, E68A, E105A, E141A) or CaM 1,2 (E32A, E68A), before Cx32, completely blocks Cx32 channel-formation, while that of CaM 3,4 (E105A, E141A) is ineffective [10]. The efficiency of CaM1,2,3,4 and CaM1,2 suggests that both CaM mutants effectively compete with wild-type CaM in interacting with CaM binding sites of Cx32. The data showing that CaM 3,4, unlike CaM 1,2, allows a virtually normal gap junction channel expression demonstrates that the Ca^2+^-activation of CaM’s N-lobe is important for gap junction formation. In addition, evidence that CaM1,2, unlike CaM 3,4, inhibits channel formation suggests that the normal Ca^2+^-affinity of the EF hand in CaM’s N-lobe is fundamental for gap junction channel formation.

Based on these and additional relevant data, we have proposed that CaM’s C-lobe interacts with a Cx32 site at resting [Ca^2+^]_i_ (~50 nM) [44]. With an increase in [Ca^2+^]_i_ above ~50 nM we think that Ca^2+^ may bind to the N-lobe allowing it to interact with a CaM-binding site of the Cx32 site and initiating the oligomerization of Cx monomers into hexameric connexons [10]. Indeed, the interaction of CaM to connexins has been shown to be important for connexin oligomerization into connexons [119]; Cx32′s assembly into hemichannels was found to be reversibly inhibited either by a synthetic calmodulin-binding peptide or by the CaM inhibitor W7 [119]. These findings were later confirmed for Cx36 as well [53]. Significantly, a recent interesting study has demonstrated that mutations at Cx43′s CaM-binding site at CT, but not at CL2, in the autosomal-dominant pleiotropic disorder called oculodentodigital dysplasia (ODDD) inhibit gap junction formation by limiting phosphorylation by Pyk2 and Src [120].” This study further confirms evidence for both a role of CaM in gap junction assembly [119] and evidence for CaM binding to CL2 and CT [42,43,44].

## 5. Chemical Sensitivity of the Fast-Vj Gate)

While we think that CaM may play a major role in the gating activity of the chemical-gate, there is no evidence that it participates in the activity of the fast-Vj gate. Indeed, the two gates are very different; as mentioned above, the channels are closed by the chemical gate slowly and completely, while the channels are closed by the fast-Vj gate rapidly but only partially. In addition, while the chemical gate always closes the channel at the negative side of Vj, perhaps because the CaM lobes have a net negative charge [70] and the channel’ pore is positively charged, the fast-Vj gating polarity is connexin dependent; rev. in [64]. The fast-Vj-gates close at the negative side of Vj in Cx32, Cx38, and Cx43 channels (negative gaters), while they close at the positive side of Vj in Cx26, Cx40, Cx50, and Cx37 channels (positive gaters). Chemical uncoupling affects negative and positive gaters in opposite ways, as acidification enhances the Vj gating sensitivity of negative gaters: Cx32 [99,102] and Cx38 [99], and inhibits that of positive gaters: Cx40 [121], Cx26 [102] Cx50 [122] and Cx37.

In this section we focus primarily on the sensitivity of the fast-Vj-gate to cytosolic acidification, without covering in detail the molecular basis of fast-Vj gating, which has been thoroughly described by other authors; for learning more thoroughly kinetics and models of fast-Vj gating, see ref. [78,79,82,83,84,85].

### 5.1. Cytosolic Acidification Inhibits the Vj-Sensitivity of the Fast-Vj Gate of Cx50 (Positive Gater) and Enhances That of the Mutant Cx50-D3N (Negative Gater)

In 2005, we questioned whether the gating polarity of the fast-Vj-gates determines the way acidification affects the Vj sensitivity of channels made of Cx50 (positive gater) (Figure 12) and that of channels made of a Cx50 mutant (Cx50-D3N) in which an aspartate, (D3), is mutated to asparagine (N3) (Figure 13) [122]. This mutation, as predicted, inverts the Cx50′s gating polarity from positive to negative.

Cx50 channels are extremely sensitive to acidification, therefore only 30% or lower CO_2_ concentrations were tested. Superfusion of oocyte pairs with solutions gasses with 30% CO_2_ significantly decreases the Vj-sensitivity of Cx50′s fast-Vj gate (Figure 12A,B) and increases that of Cx50-D3N channels (Figure 13A,B). CO_2_ also alters the kinetics of the Vj-dependent Ij’s inactivation, as it lowers the gating speed of channels made of Cx50 (Figure 12A, inset), while it increases that of channels made of Cx50-D3N (Figure 13A, inset).

The D3N mutation also increases the CO_2_ sensitivity of the chemical-gate, such that even CO_2_ as low as 5% lowers Gj (Figure 13). In Cx50 channels, Gj drops by 78% with a decrease in pH_i_ to 6.83, while in Cx50-D3N channels, Gj drops by 95% with a decrease in pH_i_ to 7.19. In controls, the mean pH_i_, was 7.73, which is close to values previously reported by us (pH_i_ = 7.63) [113] and others (pH_i_ = 7.69) [123].

Clear evidence of the inversion of fast-Vj gating polarity with the D3N mutation is demonstrated by the behavior of heterotypic Cx50-Cx50-D3N channels [122] (Figure 14). Indeed, these heterotypic channels display an asymmetric Vj behavior—with Cx50 at Vj’s positive side, the degree of Ij inactivation progressively increases with greater Vj (Figure 14A, lower traces, and Figure 14B, right side), while with Cx50 at Vj’s negative side, the inactivation of Ij is minimal (Figure 14A, upper traces, and Figure 14B, left side). The mutation D3N has shifted the polarity of Vj gating from positive (Cx50) to negative (Cx50-D3N), in such a way that with Cx50 at Vj’s positive side the fast-Vj gates of hemichannels made of both Cx50 and Cx50-D3N become active, while with Cx50 at Vj’s negative side just a few hemichannel gates become activated [122]. The slight drop in Gjss/Gjmax at negative Vj is probably due to the closure of few Cx50′s chemical gates, probably by the CaM-cork mechanism (see above). In plots of Gjss/Gjmax versus Vj, with Cx50-D3N at Vj’s the negative side (Figure 14B, right side), the Boltzmann values are V_0_ = 43.2 mV, η = 2.82, and Gjmin = 0.0035 (n = 11). It is worth noting that Gjmin is almost zero, which is drastically different from the behavior of Cx50-Cx50 channels in which the Boltzmann values are V_0_ = 24.53 mV, η = 3.3 and Gjmin = 0.15 [122].

### 5.2. Cytosolic Acidification Decreases the Vj-Sensitivity of the Fast-Vj Gate of Cx40 (Positive Gater)

Experiments on oocytes expressing homotypic Cx40 channels further proved that acidification reduces the Vj sensitivity of positive gaters (Figure 15 and Figure 16) [121]. The sensitivity of Cx40 channels to Vj was studied by means of a conventional Vj protocol (20 mV Vj steps, 120 mV maximal, 25 s duration, passed at 45 s intervals) before and during treatment with 30% CO_2_. Gj was monitored until a condition of steady state was achieved (Figure 15A). During this period, Gjpeak decreases by ~50%, while Gjss drops minimally (Figure 15A,B), such that Gjss/Gjpeak rises from 0.5 ± 0.04 to 0.82 ± 0.05 (mean ± SE, n = 5, Figure 15B). The Gjss/Gjpeak increase is clearly shown by the gradual Ij change (Figure 15A and inset) during CO_2_ treatment. After a steady state condition was achieved (Figure 15A), the CO_2_ treatment was continued for 45–60 min, so that the channels could be tested again with the conventional Vj protocol (Figure 16B,C).

When steady state conditions in 30% CO_2_ were reached, the channels manifested a significant decrease in Vj sensitivity of the fast-Vj gate (Figure 16B,C) as compared to controls (absence of CO_2_; Figure 16A,C). Plots of the relationship between Gj/Gjmax and Vj (Figure 16C), gave the following Boltzmann values: V_0_ = 36.3 mV, η = 5.4 and Gjmin = 0.21, without CO_2_ (n = 11), and V_0_ = 48.7 mV, η = 3.7 and Gjmin = 0.31, with CO_2_ (n = 9). Superfusion of 30% CO_2_ also resulted in increased speed of Vj gating. The Ij inactivation kinetics is fit by a two-term exponential function (τ1 and τ2). In the presence of 30% CO_2_, τ1 and τ2 increase. At Vj = 60 mV, τ1 rises from 1.92 ± 0.22 s to 5.99 ± 0.51 s (mean ± SE, n = 7), and τ2 rises from 0.41 ± 0.04 s to 0.87 ± 0.2 s (mean ± SE, n = 7). Both Vj sensitivity and kinetics recover to control values with extensive CO_2_ washout.

### 5.3. With Acidification the Vj-Sensitivity of the Fast-Vj gate of Cx32 (Negative Gater) Increases and That of Cx26 (Positive Gater) Decreases

In oocyte pairs expressing homotypic Cx32 (negative gater) channels, cytosolic acidification enhances Vj sensitivity (Figure 17A) [102], while in those expressing homotypic Cx26 (positive gater) channels acidification decreases Vj sensitivity (Figure 17B) [102]. The replacement of Cx32′s asparagine (2N) with aspartate (D; Cx32-N2D) changes the polarity of the fast Vj sensor to positive and cytosolic acidification decreases the sensitivity of Cx32-N2D channels to Vj (Figure 17C) as it does with Cx26 channels [102].

#### 5.3.1. Cx32

Figure 17A, shows the drop in both Gjpeak and Gjss and the enhanced Vj sensitivity of the fast-Vj gate during exposure of homotypic Cx32 channels to 100% CO_2_ [102]. Gjpeak decreases from 100% to 53.2 ± 14.4% and Gjss, from 49.9% to 18.0 ± 4.7%, such that Gjss/Gjpeak decreases from 0.950 ± 0.02 to 0.34 ± 0.02 (Figure 17A). The inset shown in Figure 17A demonstrates the Ij change, normalized to Ijpeak, with 12 s, -80 mV, Vj pulses monitored before (sweep #2) as well as during (sweep #32) uncoupling. The Gjpeak time course is biphasic, as it rises first before dropping. In contrast, the Gjss time course is a monophasic drop down to steady-state values (Figure 17A). The initial Gjpeak rise is similar to that observed with Cx45 (Figure 11), and is likely to result from activation of the Ca^2+^/CaM kinase II which may result from a small initial rise in [Ca^2+^]_i_, whereas the following greater [Ca^2+^]_i_ rise may close the channels by the chemical-gate via the Ca-CaM-cork gating mechanism. Significantly, in Cx45 the brief Gj rise is not seen with CaM-expression inhibition (Figure 11). The role of Ca^2+^/CaM kinase II in opening connexin channels has been recently reviewed [51,114].

#### 5.3.2. Cx26

Cx26 channels are more sensitive to CO_2_ than Cx32 channels, such that treatment with 100% CO_2_ virtually closes all of the channels [102]. Therefore, to activate chemical gating without closing all the channels, the oocytes were treated with 30% CO_2_ (Figure 17B). Gjpeak decreases with CO_2_ from 100% to 47.5 ± 11.6%, whereas Gjss only drops from ~41.0% to ~28.2, such that that the Gjss/Gjpeak ratio rises from 0.41 ± 0.05 to 0.59 ± 0.02 (Figure 17B). Ij normalized to Ijpeak, with Vj pulses of 12 s, −120 mV (Figure 17B, inset), sampled under control conditions (sweep #2) as well as during 30% CO_2_-induced gating (sweep #34), demonstrates the decreased Vj gating sensitivity (Figure 17B, inset). The Gjss/Gjpeak time course matches quite well those of both Gjpeak and Gjss (Figure 17B). This clearly demonstrates that Cx26 channels drastically differ from Cx32 channels (Figure 17A).

#### 5.3.3. CX32-N2D

Replacement of Cx32′s asparagine (2N) with aspartate (D) had been reported to switch the fast Vj sensor’s polarity from negative to positive [101,124]. Indeed, our experiments testing the Vj sensitivity of Cx32/Cx32-N2D channels proved that the Cx32-N2D mutant is a positive gater [102]. As predicted, the Vj sensitivity of Cx32-N2D channels exposed to 80 mV Vj pulses of 12 s duration greatly drops in the presence of CO_2_ (Figure 17C). Gjpeak decreases from 100% to ~20.2% and Gjss drops from ~59.6% to ~18.4% in the presence of 100% CO_2_, resulting in a Gjss/Gjpeak rise from 0.60 ± 0.02 to 0.91 ± 0.01 (Figure 17C). Both Gjss and Gjpeak were normalized to Gjpeak at the beginning of the experiment as 100% [102]. In Figure 17C (inset) the two current traces (12 s, −80 mV pulses), normalized to Ijpeak, clearly show the reduction in Ij inactivation caused by exposure to 100% CO_2_. Note that the Gjss/Gjpeak time course matches quite well the Gjpeak and Gjss time courses (Figure 17C), proving that Cx32-N2D channels behave like Cx26 channels (Figure 17B), while they drastically differ from Cx32 channels (Figure 17A).

## 6. Summary and Conclusions

Two types of gates are expressed in gap junction channels: a fast Vj-sensitve gate and a chemical gate. In different ways, both gates are sensitive to Vj gradients and cytosolic acidification (increased [Ca^2+^]_i_). The fast-Vj gate closes rapidly but incompletely, as it leaves the channel at a residual conductance state of 20–25% (Figure 1 and Figure 18), while the chemical-gate closes the channel slowly and totally [77] (Figure 1 and Figure 18).

The Vj behavior of the fast-Vj gate is mostly related to the NT’s charge type. In channels made of connexin/innexin with a positively charged NT, the fast-Vj gate closes at Vj’s negative side (negative gaters), whereas in channels made of connexins with negatively charged NT domain the fast-Vj gate closes at Vj’s positive side (positive gaters). In contrast, the chemical-gate closes at Vj’s negative side in all connexins/innexins channels, perhaps because the CaM lobes are negatively charged [70] and the pore (vestibule) of gap junction channels is charged positively.

Uncoupling by cytosolic acidification has an opposite effect on the gating sensitivity of the fast-Vj gate, as acidification raises the Vj sensitivity of negative gaters, such as Cx32 [99,102] and Cx38 [99] and decreases that of positive gaters, such as Cx40 [121], Cx26 [102], Cx50 [122] and Cx37. For reviews on connexin expression in vertebrate organs see [46,64,71,72,73,74,75,76].

The reason cytosolic acidification decreases the sensitivity to Vj of positive gaters and increases the Vj sensitivity of negative gaters is unclear. However, there is evidence that charged residues at the initial NT domain play a role [101]. Indeed, the NT is believed to be near the channel’s pore, and to possess the voltage sensor [101,125,126]. The general view is that at the initial portion of the NT domain, positive gaters would be so by the fact that they have an acidic residue in the second (D2) or third position (D3), while negative gaters do not (Figure 19), except in Cx43. With cytosolic acidification, the aspartic (D) residue would become protonated and the negative charge at the initial NT domain of positive gaters (Cx26, Cx37, Cx40 and Cx50; Figure 19) would be neutralized, resulting in reduced Vj sensitivity. The absence of the initial negative charge in negative gaters such as Cx32 (Figure 19), could be the reason for the increased Vj sensitivity of their fast-Vj gate during acidification; another reason could be that acidification-induced protonation of the histidine residue (H) located in the middle domain of NT increases the Vj sensitivity of Cx32′s fast-Vj gate by adding a positive charge (Figure 19). Our evidence for inversion of Vj-gating polarity in mutants Cx32-N2D [102] and Cx50-D3N [122] (Figure 19) confirms the relevance of the presence or absence of the aspartic residue (D) at the initial NT domain.

Significantly, while the fast-Vj gate closes and opens rapidly and without fluctuations (Figure 1), the chemical gate closes and opens slowly and often displays fluctuations (Figure 1B) [77]. We think that the fluctuations may reflect the behavior of a sizable gating particle, which we believe to be the N-lobe of CaM, which might be momentarily shifting in and out of the channel’s pore before assuming the final gating position.

## Figures and Tables

**Figure 1 ijms-25-00982-f001:**
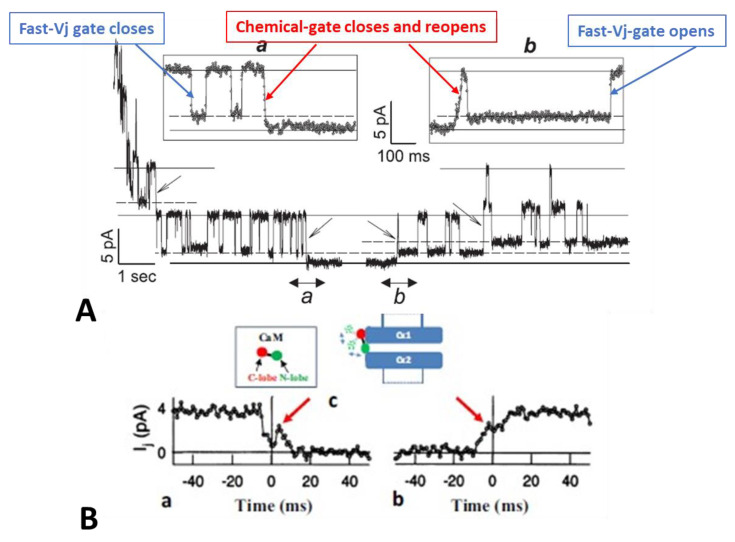
Junctional current (Ij) studied by dual whole-cell clamp (DWCC) in Cx43 transfected HeLa cells, monitored during exposure to 100% CO_2_. At a Vj of 55 mV, the fast–Vj gating of the remaining 5 open channels manifests a quick (~2 ms) Ij-flickering from open (γj main state, (**A**) solid lines) to residual (γj residual, A dashed lines) conductance, with a ratio γj (residual)/γj (main state) of 20–25% (**A**), while the chemical gate of the last open channel closes the channel totally by a slow Ij-transition (~10 ms; (**A**)a and (**B**)a). The channels reopen during recoupling by slow Ij transition (~10 ms) from closed state to open state ((**A**)b and (**B**)b). The chemical-gate’s slow transitions frequently display fluctuations during both channel closing ((**B**)a, red arrow) and opening ((**B**)b, red arrow), which may be consistent with the idea that a sizable particle, which we believe is a CaM lobe ((**B**)c), shifts momentarily in and out of the channel’s pore before assuming the final closed ((**B**)a) or open ((**B**)b) state. Adapted from Ref. [77].

**Figure 2 ijms-25-00982-f002:**
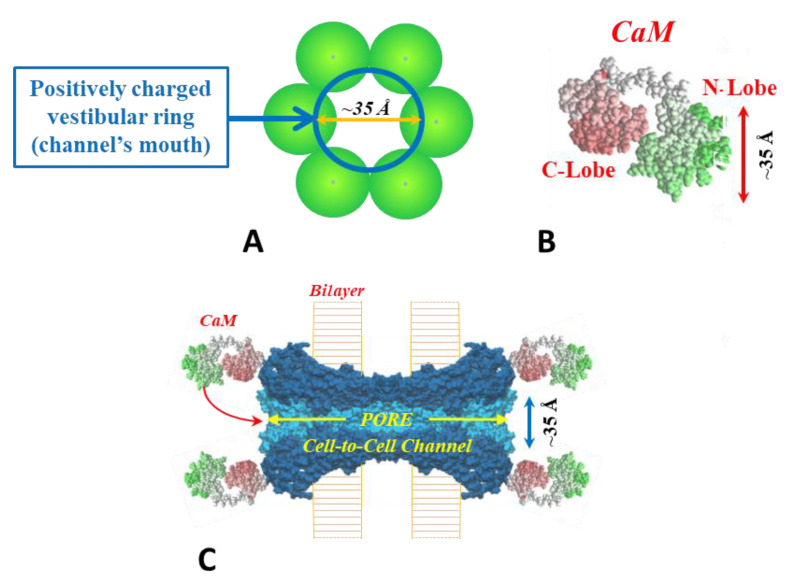
Cork-gating. The positively charged channel’s pore (**A**,**C**) and the negatively charged CaM lobes (**B**) are ~25 × 35 Å in diameter. Thus, a CaM lobe would be expected to interact and fit well within the positively charged connexon’s pore (vestibule, (**A**,**C**)). In (**C**) the channel is split to show the diameter of the channel along its length. CaM and connexon images (**B**,**C**) were given by Dr. Francesco Zonta (VIMM, University of Padua, Italy). Adapted from ref. [45].

**Figure 3 ijms-25-00982-f003:**
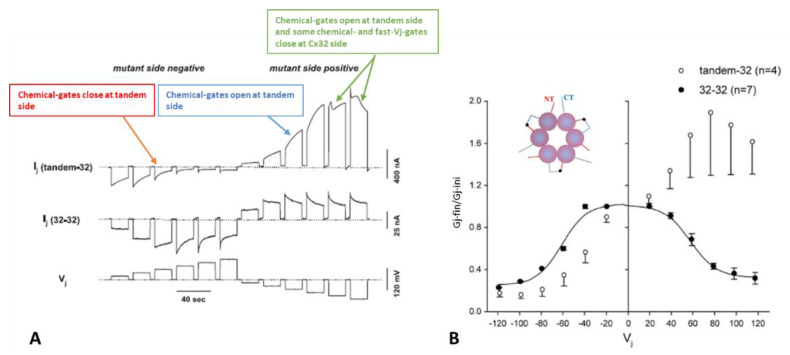
Ij response to Vj pulses in oocyte pairs expressing Cx32 channels (32–32, (**A**)) or tandem-Cx32 channels (tandem–32, (**A**)). Homotypic 32–32 channels show a conventional Vj sensitivity, as Ij decreases for Vj greater than ±40 mV (**A**). In contrast, heterotypic tandem–32 channels show an unusual Ij–Vj behavior (**A**). With negative Vj at tandem side ((**A**), left), as Vj increases both Ij initial and Ij final gradually drop to very low values, and the channels are sensitive even to Vj = 20 mV ((**A**), left). With tandem at positive Vj side ((**A**), right), Ij gradually increases from initial Ij. At Vj = 100–120 mV the behavior is more complex: as chemical-gates open at tandem side, some chemical- and fast-Vj gates close at Cx32 side. The Ij-Vj behaviors of tandem-32 and 32–32 are shown in a plot (**B**) of normalized Gj (Gj-fin/Gj-ini versus Vj). Note that in other figures the Gj–fin/Gj-ini was named Gjss/Gjmax or Gjss/Gjpk (see below). For tandem-32 channels, labeling Gj–fin/Gj-ini is more appropriate because Vj positive at tandem side increases rather that decreases Gj, at least up to Vj = 80 mV. Adapted from Ref. [69].

**Figure 4 ijms-25-00982-f004:**
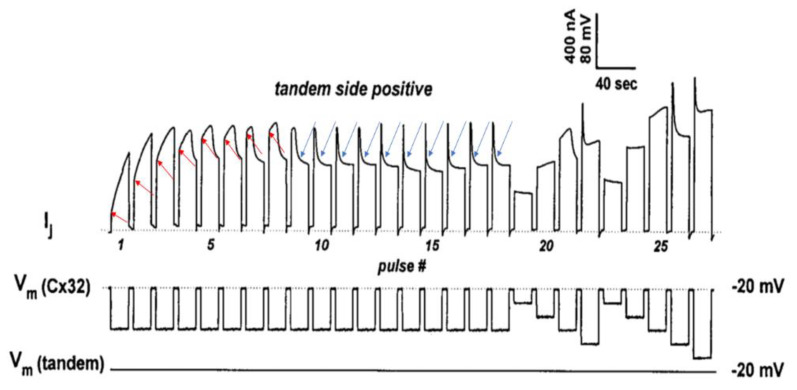
Heterotypic channels made of tandem–32. With a sequence of 60-mV Vj pulses (tandem side positive), 3 different Ij behaviors are observed. 1. Rapid Ij rise followed by slow rise in Ij (pulses #1–3). 2. Triphasic time course of Ij with initial rapid Ij rise, followed by slow Ij rise and exponential Ij decay (pulses #4–9). 3. Rapid initial Ij rise followed by exponential Ij decay to a steady-state (pulses #10–18; blue arrows). The testing of Vj protocols (tandem side positive) after the train of *V*j pulses results in a behavior of Ij–Vj like that of Cx32–Cx32 channels (pulses #19–27). In pulses #1–8 the red arrows indicate the Ij transition from rapid to slow Ij rise. The rapid Ij rise reflects the Ij of channels that were already open before the pulse; the slow rise reflects Vj-induced opening of the chemical-gate of previously closed channels. In pulses #9–18, the blue arrows indicate the closure of the fast–Vj gates at the Cx32 side. Adapted from Ref. [69].

**Figure 5 ijms-25-00982-f005:**
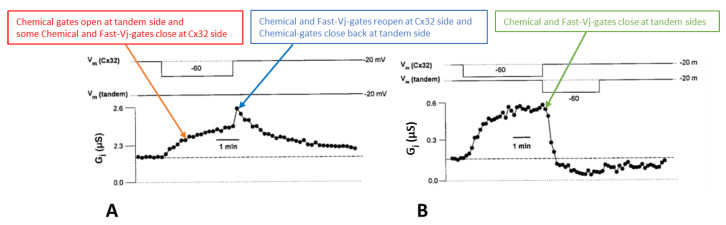
Gj response to stable Vj gradients in channels made of tandem-32. In oocytes clamped at first at Vm = –20 mV (Vj = 0 mV), the application of 40 mV Vj (positive at tandem side) progressively increases Gj by ~262% (A and B). With a return to Vj = 0 mV, Gj decreases exponentially toward the baseline (**A**). With a reversal of Vj to 40 mV (negative at tandem side, (**B**)), Gj drops exponentially to a level lower than control values (**B**), suggesting that negative Vj at tandem side effectively closes channels. Significantly, with a return to Vj = 0 mV from Vj = 40 mV, positive at tandem side, Gj rises abruptly before dropping (**A**). This reflects fast reopening of few chemical- and fast–Vj gates of the Cx32 hemichannels. Simultaneously, chemical gates start closing at the tandem side. In fact, the sudden Gj rise is not seen when Vj is reversed from tandem side positive to negative (**B**), because, while the gates of one hemichannel open those of the opposite hemichannel close. Adapted from Ref. [69].

**Figure 6 ijms-25-00982-f006:**
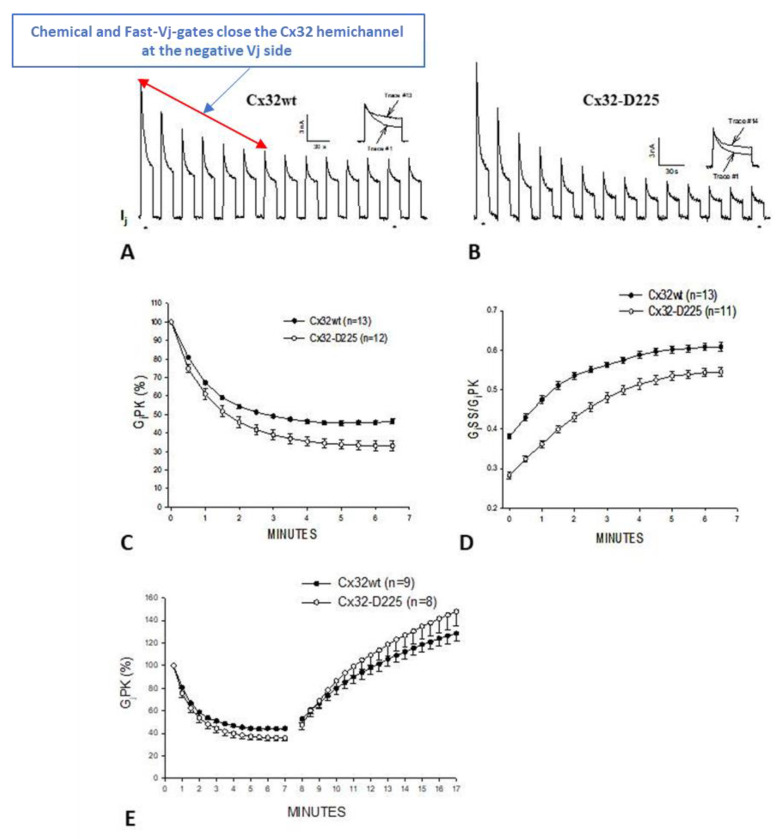
Gj and Vj-sensitivity monitored in oocyte pairs expressing Cx32 or its mutant Cx32-D225 (a mutant in which the COOH-terminal domain is deleted at residue 225). The application of trains of long Vj pulses of -100 mV Vj induces Gjpeak (GjPK) and Gj steady state (GjSS) to gradually drop by 50–60% (τ = ~1.2 min (**A**). GjPK drops more drastically (**C**), such that GjSS/GjPK rises from 0.4 to 0.6 (**D**). After the application of the Vj pulses, Gj gradually recovers with a τ = ~7 min ((**E**), minutes 8–17). More drastic results are obtained with Cx32-D225 channels (**B**–**E**). Adapted from Ref. [88].

**Figure 7 ijms-25-00982-f007:**
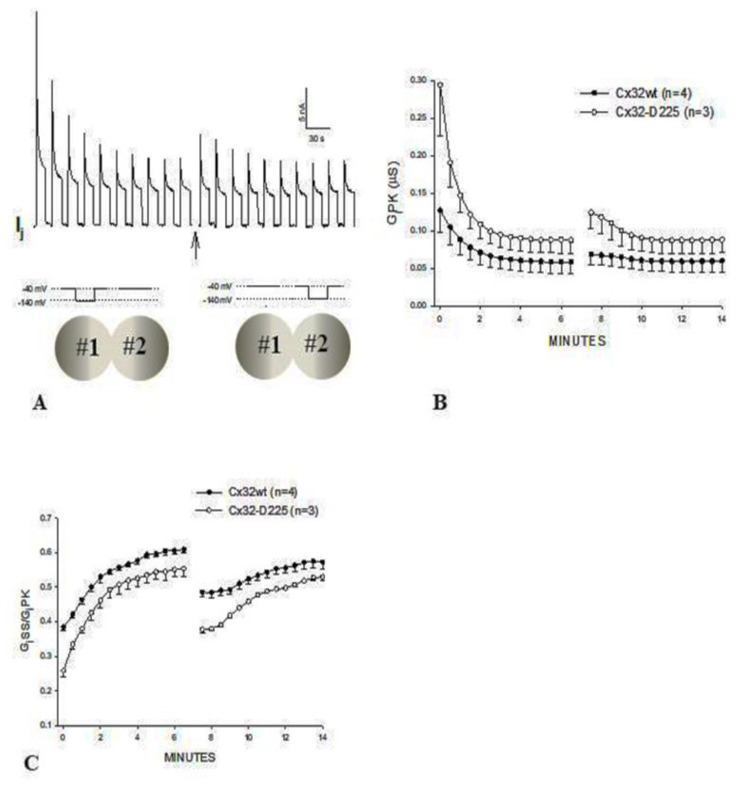
Gj and Vj-sensitivity of channels made of homotypic Cx32 or its mutant Cx32–D225. When trains of long Vj pulses (−100 mV) applied to oocyte #1 are suddenly followed by Vj pulses passed to oocyte #2 (Vj polarity reversal) (**A**), GjPK (**B**) and GjSS rise slightly before decreasing. Since fast-Vj gates and chemical gates of Cx32 close at Vj negative, with Vj polarity reversal the gates of the gated hemichannels, now exposed to Vj positive, begin to open, while those at Vj negative begin to close. GjSS/GjPK drops with a switch of Vj polarity (**C**). Even larger changes occur with Cx32-D225 (**B**,**C**). Adapted from Ref. [88].

**Figure 8 ijms-25-00982-f008:**
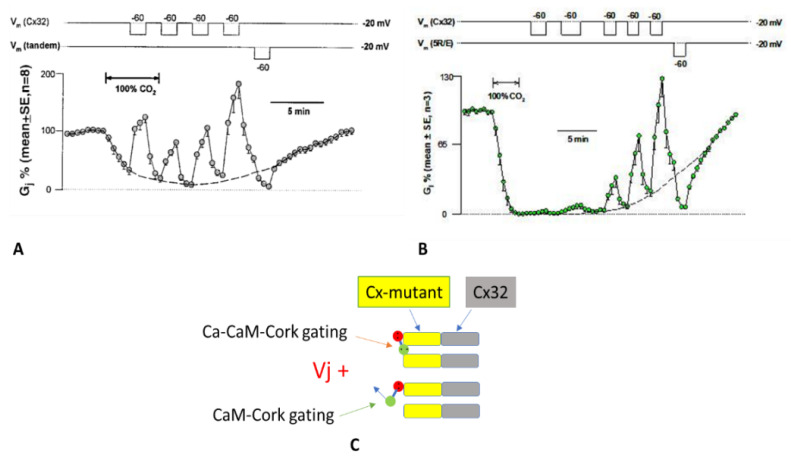
Vj effect on Gj during treatment with 100% CO_2_ in channels made of tandem–32 (**A**) or 5R/E-32 (**B**). In these mutant-Cx32 channels, Gj, greatly lowered by CO_2_ at Vj = 0 mV, reversibly rises with positive Vj positive at mutant side, while negative Vj at mutant side reversibly lowers Gj to low values (**A**,**B**). Positive Vj at mutant side is gradually less effective in raising Gj as uncoupling progresses (**A**,**B**) and gradually has greater effect during recovery (**A**,**B**). In 5R/E–32 channels Vj positive has a minimal effect at total uncoupling (**B**). The most likely reason is that there are two groups of channels: one in Ca-CaM-cork gated state (**C**, top) and the other in CaM-cork gated state (**C**, bottom). In CaM-cork gating state the gates of mutant hemichannels are opened by Vj positive (**C**, bottom), whereas in Ca–CaM-cork gating state they remain closed (**C**, top). In A and B, the dashed lines represent the predicted Gj time-course without Vj gradients. Adapted from Ref. [69].

**Figure 9 ijms-25-00982-f009:**
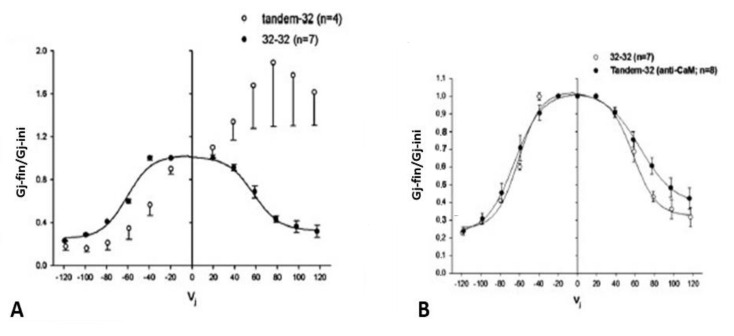
The asymmetric behavior of Ij-Vj in tandem-32 channels, which is shown in plots of normalized Gj (Gj–fin/Gj–ini) versus Vj (**A**), and presented in Figure 3, is virtually eliminated with CaM–expression inhibition ((**B**), anti–CaM). Adapted from Ref. [104].

**Figure 10 ijms-25-00982-f010:**
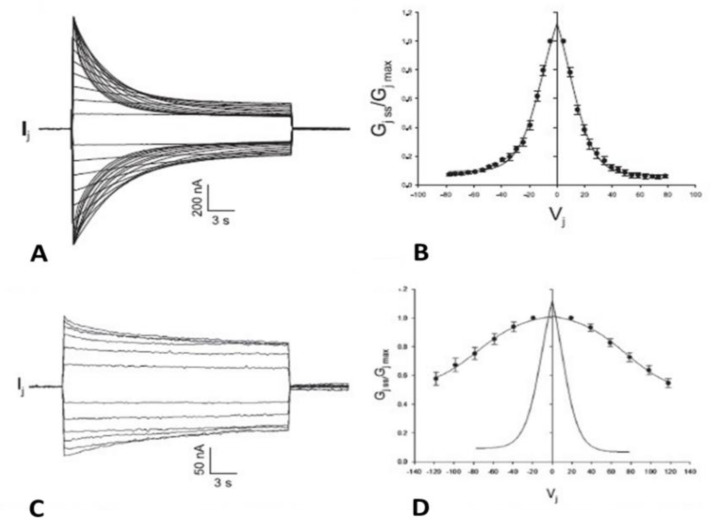
Vj sensitivity of homotypic Cx45 channels. These channels are extremely Vj–sensitive, such that Ij (**A**) and Gjss/Gjmax (**B**) are sensitive to Vj as low as ±5 mV. In Cx45 channels, Vj preferentially affects the chemical gate. CaM–expression inhibition dramatically reduces the sensitivity to Vj of Cx45 channels (**C**,**D**), to such an extent that Ij (**C**) and Gjss/Gjmax (**D**) do not drop with Vj smaller than ±40 mV. The Vj sensitivity of controls is also shown (**D**, solid line). Adapted from Ref. [110].

**Figure 11 ijms-25-00982-f011:**
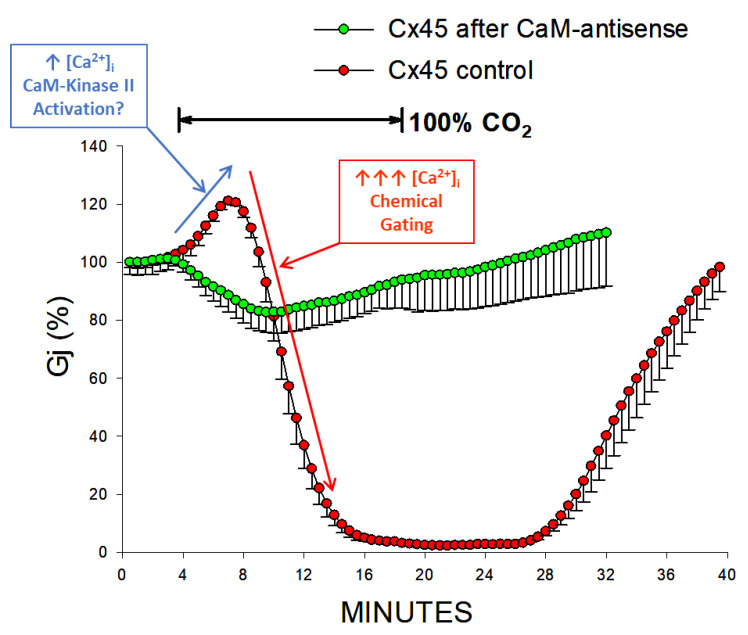
Gj of Cx45 channels exposed to 100% CO_2_. CaM-expression inhibition significantly lessens the CO_2_ sensitivity of channels made of Cx45, to such an extent that Gj reversibly drops by just ~17%, at a rate of ~3.3%/min (green circles) with 15-min exposure to 100% CO_2_. In contrast, in controls, Gj decreases to 0% (red circles). The Gj rise that precedes its drop in controls is not seen with CaM-expression inhibition. The initial Gj rise probably results from activation of Ca^2+^/CaM kinase II (blue arrow), while the following larger [Ca^2+^]_i_ rise (red arrow) causes channel gating by activation of the chemical gate. Adapted from Ref. [110].

**Figure 12 ijms-25-00982-f012:**
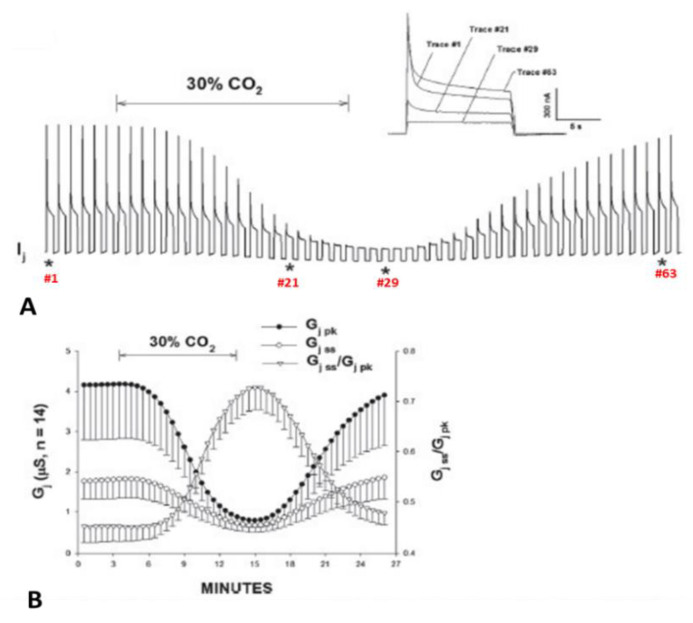
Gj of Cx50 channels exposed to 30% CO_2_. Acidification of the cytosol results in cell–cell uncoupling and largely reduces the Vj sensitivity of positive gaters as channels made of Cx50 (**A**,**B**) or Cx26 (see below). This is proven by the great GjSS/GjPK rise (**B**). The asterisks in A indicate the number of the traces shown in A (inset). Adapted from Ref. [122].

**Figure 13 ijms-25-00982-f013:**
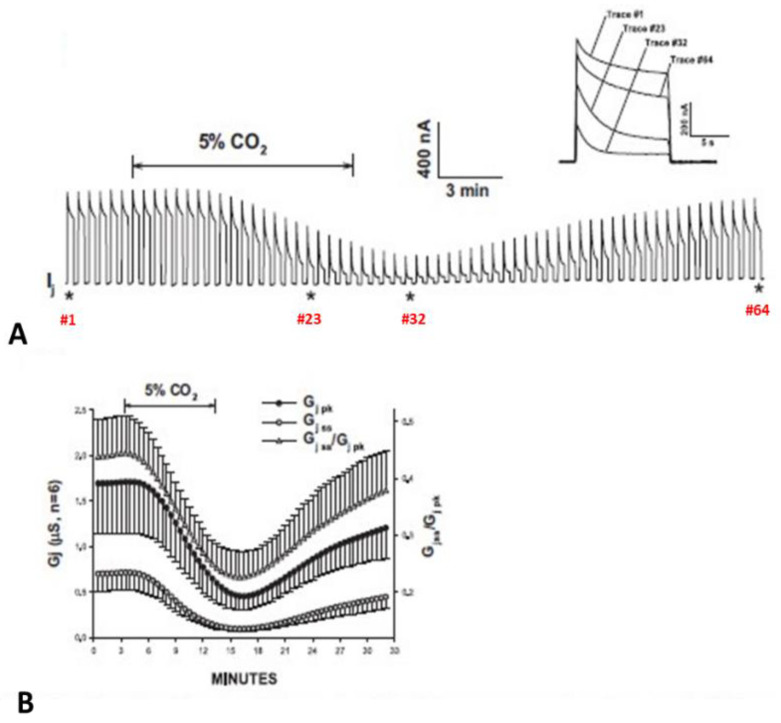
Gj of homotypic Cx50-D3N channels exposed to 5% CO_2_. CO_2_-induced cytosolic acidification causes uncoupling even with such a minimal percentage of CO_2_. The D3N mutation inverts Cx50′s gating polarity from positive to negative. Predictably, this mutation dramatically increases the fast-Vj gate’s Vj sensitivity (**A**,**B**). Note that the GjSS/GjPK ratio greatly drops with exposure to CO_2_ (**B**). The asterisks in A indicate the number of the traces shown in A (inset). Adapted from Ref. [122].

**Figure 14 ijms-25-00982-f014:**
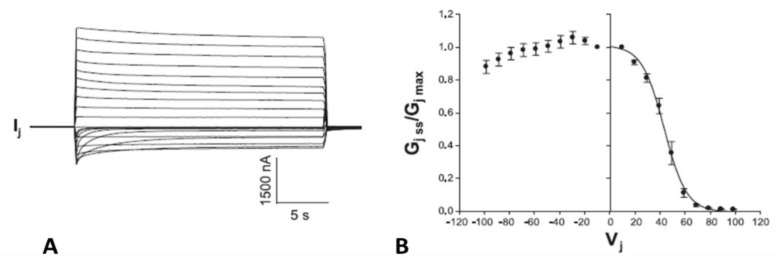
Vj sensitivity of heterotypic Cx50/Cx50-D3N channels. With Cx50 at Vj-positive side, the Ij inactivation progressively rises with Vj (**A**, bottom traces), while with Cx50 at V-negative side, Ij inactivation is minimal (**A**, top traces). This asymmetric Vj behavior, consistent with Cx50–D3N’s Vj-polarity inversion from positive to negative, is clearly seen in the relation between Vj and normalized Gj (Gjss/Gjmax), plotted with the sign of Vj relative to Cx50 hemichannels (**B**). The small Cjss/Gjmax drop with negative Vj at the Cx50 side (**B**, left side) is probably caused by the gating of a few Cx50 hemichannels by the chemical gate (see Figure 6 and Figure 7 for Cx32). Adapted from Ref. [122].

**Figure 15 ijms-25-00982-f015:**
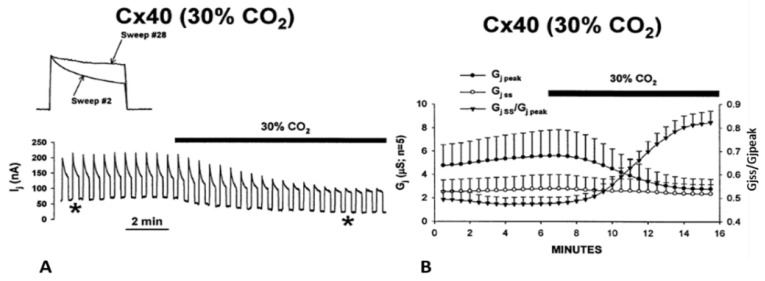
Vj sensitivity of Cx40 channels exposed to 30% CO_2_ (Cx40 is a positive gater). Superfusion with 30% CO_2_ (**A**,**B**) causes Gjpeak to drop by ~50%, while Gjss decreases minimally (**A**. inset, and **B**), such that Gjss/Gjpeak increases from 0.5 ± 0.04 to 0.82 ± 0.05 (mean ± SE, n = 5, **B**). The Gjss/Gjpeak ratio demonstrates a large drop in Vj-sensitivity of the fast-Vj gate. The asterisks in A indicate the numbers of traces shown in A (inset). Adapted from Ref. [121].

**Figure 16 ijms-25-00982-f016:**
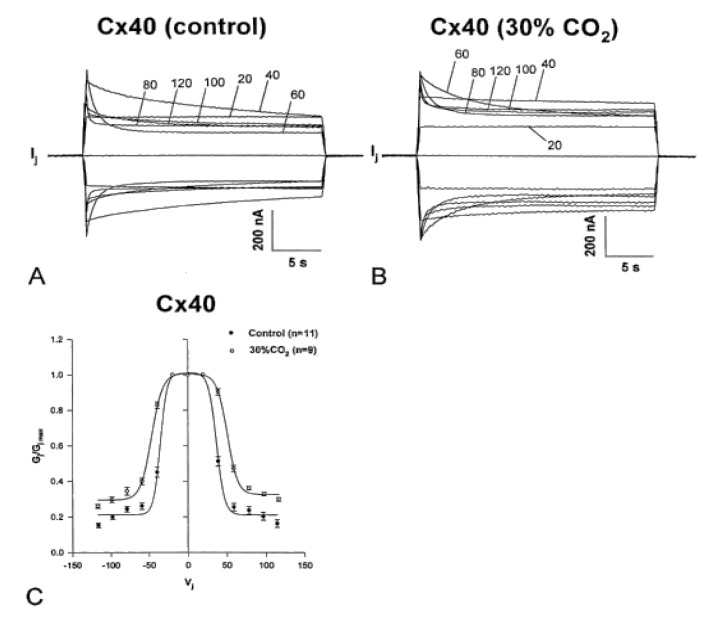
Vj sensitivity and kinetics of homotypic channels made of Cx40 (positive gater) exposed to 30% CO_2_. The oocytes were studied by applying the conventional Vj protocol before (**A**,**C**) and during (**B**,**C**) treatment with 30% CO_2_. Cytosolic acidification with 30% CO_2_ causes a significant drop in fast-Vj gate’s Vj sensitivity (**B**,**C**), as compared to controls (absence of CO_2_; (**A**,**C**)). In the relationship between Gj/Gjmax and Vj (**C**), the Boltzmann values are V_0_ = 36.3 mV, η = 5.4 and Gjmin = 0.21 without CO_2_ (n = 11), and V_0_ = 48.7 mV, η = 3.7 and Gjmin = 0.31, with CO_2_ (n = 9). The Ij relaxation curves seen in (**A**,**B**) are labeled with the corresponding values of Vj. Adapted from Ref. [121].

**Figure 17 ijms-25-00982-f017:**
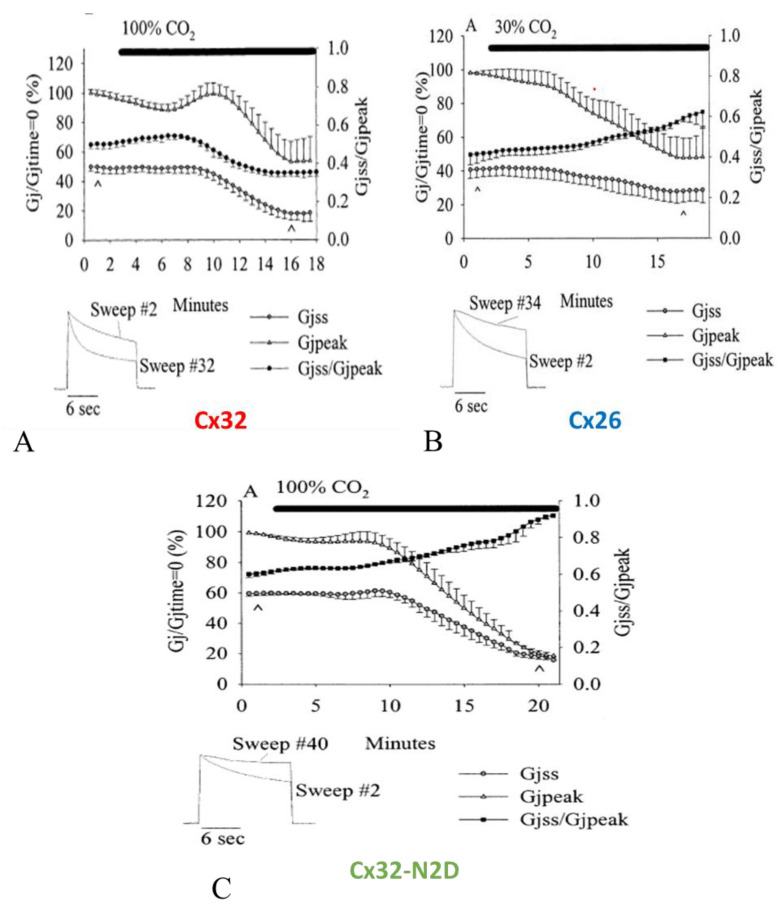
Changes in Gjpeak, Gjss and Gjss/Gjpeak in homotypic Cx32, Cx26 and Cx32-N2D channels before and during CO_2_ treatment (**A**). Both Gjpeak and Gjss (triangles and circles, respectively) of Cx32 channels drop during uncoupling, such that Gjss/Gjpeak (dots) decreases, reflecting an increased Vj sensitivity of the fast-Vj gate. (**B**). Gj monitored in Cx26 channels during exposure to 30% CO_2_; Gjpeak and Gjss decrease with CO_2_, such that Gjss/Gjpeak (dots) increases, reflecting a decreased Vj sensitivity of the fast-Vj gate. (**C**). Gj changes in Cx32-N2D channels treated with 100% CO_2_; Gjpeak and Gjss decrease with CO_2_, such that Gjss/Gjpeak (squares) increases, reflecting a decreased Vj sensitivity of the fast-Vj gate. Data are mean ± SEM. The insets in (**A**–**C**) show Ij traces (sweeps) normalized to Ijpeak in control conditions and with exposure to CO_2_ (the data points are labelled with ^). Adapted from Ref. [102].

**Figure 18 ijms-25-00982-f018:**
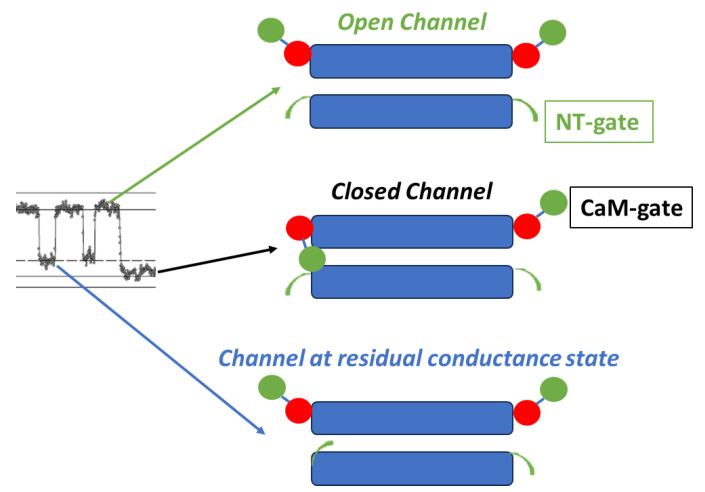
Schematic representation of the two gates. The chemical gate (CaM-gate?) closes slowly and completely, while the fast-Vj gate closes quickly and partially, maintaining it in residual conductance state. The image on the left shows the closure of both channel gates, monitored at single-channel level by dual whole-cell clamp in HeLa cells transfected with Cx43, during exposure to 100% CO_2_ (see Figure 1) The red and green colors in CaM indicate C-lobe and N-lobe, respectively.

**Figure 19 ijms-25-00982-f019:**
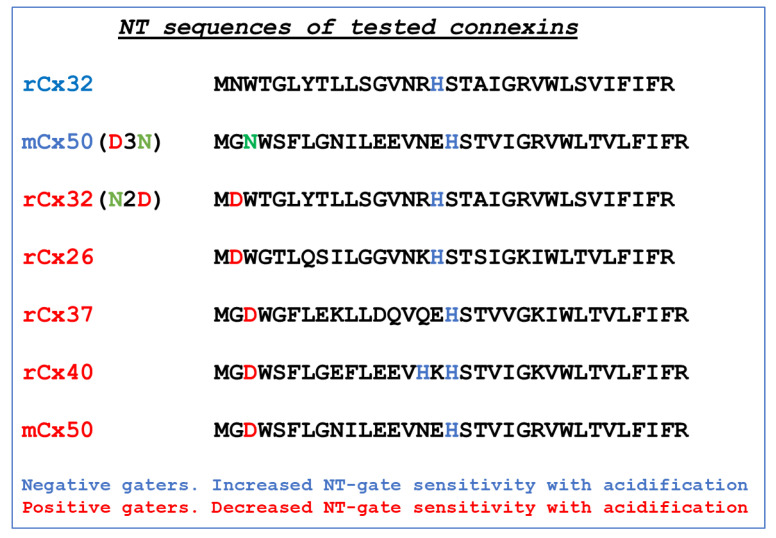
Amino acid sequences of connexins’ NT-domain. Note the presence of an aspartic (D) residue in Cx26, Cx37, Cx40, Cx50 and Cx32-N2D and its absence in Cx32 and Cx50-D3N. All of them contain a histidine (H) residue in the middle of the domain.

## Data Availability

Data contained within the article.

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
