# Peer review of "Gap Junction Channel Regulation: A Tale of Two Gates—Voltage Sensitivity of the Chemical Gate and Chemical Sensitivity of the Fast Voltage Gate"

_ijms, 2024, doi:10.3390/ijms25020982_

Round 1

Reviewer 1 Report

Comments and Suggestions for Authors

For several decades, it has been appreciated that gap junction channels have distinct gates. One is primarily affected by voltage imposed across the gap junction channel (transjunctional voltage, Vj) and the other is affected by chemicals, including intracellular H ions.  The author of this review has long argued that closure of the chemical gate is in fact pore-plugging by the Ca sensitive molecule calmodulin and here goes so far as to preposterously term it the CaM gate.    The review is an interpretation of previous data from his laboratory in this context and is of historical interest.  However, it is an opinion piece that selects the studies cited to support the thesis, rather than a balanced view summarizing current status of knowledge of gap junction biophysics.  It should be labeled as an essay and ideally paired with a contrary viewpoint.

The origin of the belief that Ca closes gap junction channels is traced to studies in which damaged myocardium regained its integrity after injury, and recovery was less complete when Ca was absent from the medium.  Injury-induced loss of intercellular communication takes time, and the extent to which “healing-over” is due to channel closure as distinct from retrieval from the membrane is not well established.  Moreover, precise measurement and selective manipulation of calcium ions within nanoscopic subcellular compartments is trickier than with H ions and other stimuli.  Thus, many studies that have appeared to distinguish among hypotheses have generated ambiguous results.

In the opinion of this reviewer, it has been well established that the calcium sensitive molecule calmodulin binds to certain connexins and that some connexins are phosphorylated by Ca dependent CaM activation of its kinase.  Whether Ca itself or Ca dependent binding of CaM to a connexin directly results in channel opening or closure is not established, and in some cases the consequence of intracellular Ca elevation is stronger, rather than weaker intercellular coupling (one prominent example being in neurons, where facilitation is achieved).  By contrast, the effects of intracellular acidification, when both junctional conductance and intracellular pH are simultaneously measured, seem well attributed to titration of connexin residues resulting in rapid, reversible channel closure.  Evidence is strong that each connexin forms channels with distinct pH sensitivity, as quantified by apparent pK values, and deletion or substitution of amino acid residues changes sensitivity, in some cases without changing Ca sensitivity.   

Author Response

ANSWERS TO REVIEWER #1

The author is grateful for the helpful comments of the reviewer and has modified the paper as closely as possible to his/her comments.

For several decades, it has been appreciated that gap junction channels have distinct gates. One is primarily affected by voltage imposed across the gap junction channel (transjunctional voltage, Vj) and the other is affected by chemicals, including intracellular H ions.  The author of this review has long argued that closure of the chemical gate is in fact pore-plugging by the Ca sensitive molecule calmodulin and here goes so far as to preposterously term it the CaM gate. The review is an interpretation of previous data from his laboratory in this context and is of historical interest.  However, it is an opinion piece that selects the studies cited to support the thesis, rather than a balanced view summarizing current status of knowledge of gap junction biophysics.  It should be labeled as an essay and ideally paired with a contrary viewpoint.

I agree that our review has focused primarily on the role of Ca-CaM in gating. But the major focus of the review is not on the role of calcium versus pH in gating, but rather on the voltage sensitivity of the chemical gate. Regardless, we agree with the reviewer that mentioning of the numerous studies of the role of calcium and pH in gating needs to be addressed. Therefore, we now cite both evidence for as direct action of H+ on hemichannels and a thorough discussion on the effects of Ca and pH on gating that we have published in a book in 2019. evidence for a direct role of H+ in gating has been reported in connexin46 hemichannels (Trexler et al., 1999. For a thorough discussion on the role of H+i in connexin channel gating see Chapter #4 of our book (2019).

The origin of the belief that Ca closes gap junction channels is traced to studies in which damaged myocardium regained its integrity after injury, and recovery was less complete when Ca was absent from the medium.  Injury-induced loss of intercellular communication takes time, and the extent to which “healing-over” is due to channel closure as distinct from retrieval from the membrane is not well established.  Moreover, precise measurement and selective manipulation of calcium ions within nanoscopic subcellular compartments is trickier than with H ions and other stimuli.  Thus, many studies that have appeared to distinguish among hypotheses have generated ambiguous results.

With do respect, I would like to remind the reviewer that evidence for the effectiveness of high nanomolar to low micromolar [Ca2+]i in gating has been widely reported. See below a summary written in Chapter  6.1: ”[Ca2+]i effective on Channel Gating” published in our of our earlier review review published in IJMS titled “Calmodulin-Mediated Regulation of Gap Junction Channels” 2020, 21 (2), 485.

  1. Chemical gating is sensitive to [Ca2+]i in the nM range [4,9,10,80]. Since connexins do not have sequences capable of binding Ca2+ in the nM range, gating must be mediated by a CaM-like protein—CaM being the most obvious;
  2. CaM binds to connexins [78,79,107,108,127,156];
  3. Most connexins have a CaM binding site at NT, CL2 and CT1 domains (Figure 17B) [9,10,80,133]. Most relevant for gating are likely to be the CL2 (Figure 11) and NT sites [9,55,80,111,112,127,128,133];
  4. Peptides mimicking the CaM-binding sites of various connexins bind CaM with high affinity [9,55,80,111,112,127,128,133]. Recent data show that in several connexins CaM binds to the CL2 site both in Ca2+-dependent and -independent ways [111,112], suggesting that CaM is anchored to connexins at resting [Ca2+]i;
  5. CaM and connexins co-localize at gap junctions and intracellular spots [107,108,114,126,127];
  6. Each of the two negatively charged CaM’s lobes is ~25 × 35 Å in size [176], which is the same size as the positively charged channel’s mouth (vestibule; Figure 18) [177–179];
  7. Chemical gating is eliminated by the inhibition of CaM expression [73,105,106].
  8. CaM blockers (inhibitors) prevent uncoupling by acidification and/or increased [Ca2+]i [4,38,46,76,77,93,94,96–98,122];
  9. Overexpression of CaMCC, a CaM mutant with a higher Ca2+-affinity, greatly increases the chemical gating sensitivity of Cx32 channels [107,108];
  10. At the single channel level, the chemical/slow gate opens and closes completely and very slowly, and most often displays fluctuations (Figure 12) [161], consistent with the idea that a large particle, likely to be a CaM lobe, flickers in and out of the channel’s mouth before settling in the final position;
  11. Channels made of human–Cx40, a connexin that lacks the CL2’s CaM-binding site (Figure 11), are not gated by increased [Ca2+]i [38]. In contrast, in channels made of rat-Cx40, which has the CL2’s CaM-binding site (Figure 11), chemical gating is fully functional [117];
  12. An increase in [Ca2+]i opens Cx32 and Cx43 hemichannels [103,104,153]; this is prevented by W7 [104], suggesting a CaM role in hemichannel gating opposite to that in cell-to-cell channels. The direct CaM role in hemichannel gating was also reported for Cx50 [125] and Cx46 [156] channels.

In the opinion of this reviewer, it has been well established that the calcium sensitive molecule calmodulin binds to certain connexins and that some connexins are phosphorylated by Ca dependent CaM activation of its kinase.  Whether Ca itself or Ca dependent binding of CaM to a connexin directly results in channel opening or closure is not established, and in some cases the consequence of intracellular Ca elevation is stronger, rather than weaker intercellular coupling (one prominent example being in neurons, where facilitation is achieved).

I agree with the reviewer that calcium is likely to have more than one effect on gating. Indeed, as we suggested in regard of the effect of CO2 on Cx45 and Cx32 (see below) it is likely that a very low increase in [Ca2+]i, above resting values increases cell-cell coupling, while a larger increase closes the channels. The role of Ca2+/CaM kinase II in opening connexin channels has been recently reviewed by Zoidl and Spray [Zoidl, 2021 #954]. This reference has been added regarding Figures 11 and 17.

The brief increase in Gj that precedes the Gj drop in controls (Figure 11) is absent with inhibition of CaM expression; the initial Gj rise may result from Ca2+/CaM kinase II activation by a small rise in [Ca2+]i, while the subsequent larger [Ca2+]i rise may close the channels by activating the Ca-CaM-Cork gating (Figure 11).

The time course of Gjpeak is biphasic, as it increases first before dropping, while that of Gjss is a monophasic drop to steady-state values (Figure 17, A). The initial rise in Gjpeak is similar to that seen in Cx45 (Figure 11), and may result from Ca2+/CaM kinase II activation by an initial small rise in [Ca2+]i, while the subsequent larger increase in [Ca2+]i may close the channels by activating the Ca-CaM-gate.

 By contrast, the effects of intracellular acidification, when both junctional conductance and intracellular pH are simultaneously measured, seem well attributed to titration of connexin residues resulting in rapid, reversible channel closure. With due respect, this has not been convincingly proven.  Evidence is strong that each connexin forms channels with distinct pH sensitivity, as quantified by apparent pK values, and deletion or substitution of amino acid residues changes sensitivity, in some cases without changing Ca sensitivity. But this does not prove a direct effect of H+ on connexins.   

Submission Date

Reviewer 2 Report

Comments and Suggestions for Authors

In this well planned and quite clearly illustrated review, Professor Peracchia summarizes both classical and current evidence for the two major mechanisms that regulate both slow and fast opening and closing intercellular channels that are regulated by a variety of different gap junction or connexin proteins.  The selected material clearly sets out known differences between fast and slow gating often referred to as voltage-dependent or chemical-dependent gating.  The essential function of intercellular communication provides a firm basis for interest in the core material that is presented.  Given that this is a review, I have a number of general questions and some specific suggestions that can be addressed with editing and rewriting and perhaps additions to the Table that has been provided (Figure 19).  These include:

1.  In a number of different sections but particularly in the last third of the review beginning at about line 500 gap junction coupling is altered by application of very high levels of CO2.  These maneuvers, either 30% or 100%, would be expected to be toxic but the resulting changes are presented and discussed in terms of physiological gaiting as a result of intracellular acidification.  Please provide information concerning the level of acidification and comments concerning levels of changes in ambient CO2 that are known to occur in physiological or pathophysiological settings.

2.  After reading the material provided, I wonder whether the authors could comment or speculate concerning if the fast mode of gating subserves physiological intercellular communication in mammalian syncytia;  and the slower mode of gating offers intercellular transfer of metabolites or second messengers?  Even a partial or preliminary consideration of this would offer a functional context.

3.  The authors make appropriate and often clever use of heterologous expression of various isoforms of connexin proteins.  It would be useful and informative to indicate which mammalian systems predominantly express e.g., connexin 32, 40, 45 and/or 43.  This could perhaps be done by altering Figure 19 so that it becomes a Table.

4.  The text of this review is logically planned and, in general, quite clearly presented.  However, there are exceptions and I would recommend:

a)  Rewording and simplifying the title

b) Rewriting to shorten and clarify the Abstract

c)  Rewriting the sentence found at lines 191 through 193 to clarify this point

5.  Professor Peracchia is to be commended for his consistent high quality and in fact transformative body of work concerning the gaiing of gap junctional channels.  Nonetheless, this review makes use of and is based on a total of 111 References, 28 of which are from the Peracchia group.  Indeed four or five Peracchia References provide much of the essential data and rationale.  I am not suggesting that this is excessive self-citation but it would perhaps be useful to include a statement indicating that the majority of the material and Figures is from one research group and provide a brief rationale.

Minor Points

As mentioned, this is a current, high quality review article.  However, certain choices of words or stylistic English phrases could perhaps be reconsidered and improved.  Examples include: 

Line 65 - 'we feel confident to believe'

Line 89 - 'but we believe'

Line 107 - 'such as in poorly coupled cell pairs'

Line 133 - 'channels possess'

Line 140 - 'for making easier to understand'

Line 178 -  In this line (and in a number of Figure Legends) 'channel's mouth'

Line 643 and elsewhere - 'the idea that'

Line 658 - 'we feel that'

Legend of Figure 3 - 'see in the following'

Legend of Figure 5 - 'closing back'

Legend of Figure 8 - 'the reason is that we are dealing with'

Legend of Figure 13 - 'cytosolic acidification'.  This is not shown.

Legend of Figure 16 - 'demonstrates' should perhaps be 'reveals'

Legend of Figure 17 - the maneuver 'sweep' needs to be defined

Line 643 - 'the idea is that' 

Line 649 - 'in our case'

Line 653 - 'may confirm'

Line 658 - 'we feel'

Line 660 - 'this idea'

Comments on the Quality of English Language

This is a well planned, extensively illustrated and quite clearly presented review.  Suggestions for rewording or clarifications are provided above in Comments to Author.

Author Response

ANSWERS TO REVIEWER #2

The author is grateful for the helpful comments of the reviewer and has modified the paper as closely as possible to his/her comments.

Comments and Suggestions for Authors

In this well planned and quite clearly illustrated review, Professor Peracchia summarizes both classical and current evidence for the two major mechanisms that regulate both slow and fast opening and closing intercellular channels that are regulated by a variety of different gap junction or connexin proteins.  The selected material clearly sets out known differences between fast and slow gating often referred to as voltage-dependent or chemical-dependent gating.  The essential function of intercellular communication provides a firm basis for interest in the core material that is presented.  Given that this is a review, I have a number of general questions and some specific suggestions that can be addressed with editing and rewriting and perhaps additions to the Table that has been provided (Figure 19).  These include:

  1. In a number of different sections but particularly in the last third of the review beginning at about line 500 gap junction coupling is altered by application of very high levels of CO These maneuvers, either 30% or 100%, would be expected to be toxic but the resulting changes are presented and discussed in terms of physiological gaiting as a result of intracellular acidification.  Please provide information concerning the level of acidification and comments concerning levels of changes in ambient CO2 that are known to occur in physiological or pathophysiological settings.

Yes. 100% CO2 is much greater than the physiological range, but it is the standard way to test cell-cell uncoupling. Since the work of Turin and Warner (1977) it is has been customary for all researchers to use this %. Indeed, in the body PaCO2 ranges physiologically between 38- and 45-mm Hg (4.7-6.0 kPa) and PvCO2 is ~ 47 mmHg. The use of CO2 by gassing the bathing solution with 30-100% CO2 is the conventional method to test channel gating. In our measurements, 100% CO2 causes a drop in pHi to ~ 6.3, like values published by others.

  1. After reading the material provided, I wonder whether the authors could comment or speculate concerning if the fast mode of gating subserves physiological intercellular communication in mammalian syncytia; and the slower mode of gating offers intercellular transfer of metabolites or second messengers?  Even a partial or preliminary consideration of this would offer a functional context.

In syncytia, the cells are communicating widely via plasma membrane fusion that creates large pores. Therefore, it is irrelevant in terms of cell-cell communication whether gap junction channels of syncytia are in a gated state, with both fast and slow gates in closed state.

  1. The authors make appropriate and often clever use of heterologous expression of various isoforms of connexin proteins.  It would be useful and informative to indicate which mammalian systems predominantly express e.g., connexin 32, 40, 45 and/or 43.  This could perhaps be done by altering Figure 19 so that it becomes a Table.

The following has been added

These connexins are primarily expressed in the following tissues:

Cx32: hepatocytes, exocrine pancreas, kidney, myelinated Schwann cells.

Cx38: Xenopus cells.

Cx43: heart ventricular and atrial cardiomyocytes, endothelial cells, smooth muscle, and fibroblasts.

Cx26: liver.

Cx40: cardiomyocytes of atria and cardiac conduction system (His bundle and upper bundle branches).

Cx50: lens epithelium, retina (horizontal cells and Müller cells), spinal cord tissue.

Cx37: brain, uterus, ovary, endothelial cells of blood vessels.

For thorough review articles on connexin expression in vertebrate organs see [36, 51, 57-62].

  1. The text of this review is logically planned and, in general, quite clearly presented.  However, there are exceptions and I would recommend:
  2. a)  Rewording and simplifying the title

Done

  1. b) Rewriting to shorten and clarify the Abstract

Done

  1. c)  Rewriting the sentence found at lines 191 through 193 to clarify this point

Done

  1. Professor Peracchia is to be commended for his consistent high quality and in fact transformative body of work concerning the gaiing of gap junctional channels.  Nonetheless, this review makes use of and is based on a total of 111 References, 28 of which are from the Peracchia group.  Indeed four or five Peracchia References provide much of the essential data and rationale.  I am not suggesting that this is excessive self-citation but it would perhaps be useful to include a statement indicating that the majority of the material and Figures is from one research group and provide a brief rationale.

The number of references with the name “Peracchia” has been reduced from ~28% to 15.7%. I wish I could do more, but it is impossible. Sorry! The reviewer should realize that I worked in this field for the past 57 years and much of the work on the effect of calcium and calmodulin came from our lab, as many of the established gap junction researchers refused to consider the relevance of calcium-calmodulin.

Minor Points

As mentioned, this is a current, high quality review article.  However, certain choices of words or stylistic English phrases could perhaps be reconsidered and improved.  Examples include: 

Line 65 - 'we feel confident to believe'

Done

Line 89 - 'but we believe'

Done

Line 107 - 'such as in poorly coupled cell pairs'

Done

Line 133 - 'channels possess'

Done

Line 140 - 'for making easier to understand'

Done

Line 178 -  In this line (and in a number of Figure Legends) 'channel's mouth'

Replaced with “pore”

Line 643 and elsewhere - 'the idea that'

Done

Line 658 - 'we feel that'

Done

Legend of Figure 3 - 'see in the following'

Changed to “see below”

Legend of Figure 5 - 'closing back'

Done

Legend of Figure 8 - 'the reason is that we are dealing with'

Done

Legend of Figure 13 - 'cytosolic acidification'.  This is not shown.

The application of CO2 obviously causes acidification

Legend of Figure 16 - 'demonstrates' should perhaps be 'reveals'

Done

Legend of Figure 17 - the maneuver 'sweep' needs to be defined

Done

Line 643 - 'the idea is that' 

Done

Line 649 - 'in our case'

Done

Line 653 - 'may confirm'

Done

Line 658 - 'we feel'

Done

Line 660 - 'this idea'

Done

Comments on the Quality of English Language

This is a well planned, extensively illustrated and quite clearly presented review.  Suggestions for rewording or clarifications are provided above in Comments to Author.

Round 2

Reviewer 1 Report

Comments and Suggestions for Authors

It is now generally accepted by gap junction biophysicists that open probability of these channels is influenced by transjunctional voltage and, rather independently, by a “chemical” gate that is variably sensitive to Ca, pH, lipophiles and perhaps other drugs such as carbenoxolone and mefloquine. This review presents itself as an overview of the sensitivities of each gate to the other class of stimuli (how chemicals affect Vj gating and how voltage affects chemical gating).  Although the author gets across certain insights about channel operation from analysis of pH effects on Vj sensitivity, the sections dealing with the chemical gate repeat provide an essay in which the author’s previously published data are selectively interpreted and it must be labeled as such.  Major concerns remain:

The term “CaM-gate” is preposterous.  Not only is a direct effect of Ca on GJ channels not firmly established, the author agrees that the channel itself does not have substantial Ca sensitivity and the “gating” is instead hypothesized to be through plugging the channel by a separate molecule, rather than closing a gate.  But there is no conclusive evidence that CaM is required for chemical closure of any gap junction channel. And nowhere are other gating agents mentioned, such as the highly potent lipophilic volatile anesthetics and long-chain alcohols whose lack of strong effects on the Vj gate enabled the earliest single channel recordings for most connexins.  Additionally, the term “NT-gate” is also not preferred over the generally used “Vj gate”, as it assigns a single domain to the channel closure which more likely involves interactions of amino acids in several portions of molecule.

nM sensitivity to Ca ions is an exaggeration; reports of effects below 100 nM are dubious, and most studies on hemichannels consider mM levels!

The sections dealing with modulation of Vj gating by chemical agents is somewhat novel and useful, but the section on the voltage sensitivity of the chemical gate and involvement of CaM are well covered territory by that author, including many reviews and even a full book.

Author Response

It is now generally accepted by gap junction biophysicists that open probability of these channels is influenced by transjunctional voltage and, rather independently, by a “chemical” gate that is variably sensitive to Ca, pH, lipophiles and perhaps other drugs such as carbenoxolone and mefloquine. This review presents itself as an overview of the sensitivities of each gate to the other class of stimuli (how chemicals affect Vj gating and how voltage affects chemical gating).  Although the author gets across certain insights about channel operation from analysis of pH effects on Vj sensitivity, the sections dealing with the chemical gate repeat provide an essay in which the author’s previously published data are selectively interpreted and it must be labeled as such.  Major concerns remain:

The term “CaM-gate” is preposterous. Agree! Accordingly, I have replaced both “CaM-gate” and NT-gate” with “chemical gate” and fast-Vj gate”.

Not only is a direct effect of Ca on GJ channels not firmly established the author agrees that the channel itself does not have substantial Ca sensitivity and the “gating” is instead hypothesized to be through plugging the channel by a separate molecule, rather than closing a gate.  But there is no conclusive evidence that CaM is required for chemical closure of any gap junction channel. Agree! This sentence has been added: “While we feel that there is strong evidence for a Ca-CaM role in chemical gating, it should be stressed that the CaM-based gating mechanism is yet to be unequivocally proven”.

And nowhere are other gating agents mentioned, such as the highly potent lipophilic volatile anesthetics and long-chain alcohols whose lack of strong effects on the Vj gate enabled the earliest single channel recordings for most connexins

Agree! This sentence has been added: “In addition to Ca-CaM, there is evidence that the channels can be closed by lipophilic volatile anesthetics and long chain alcohols [45-48]

Also added: “Indeed, the use of anesthetics enabled the earliest single channel recordings for most connexins [49-51]”.

  Additionally, the term “NT-gate” is also not preferred over the generally used “Vj gate”, as it assigns a single domain to the channel closure which more likely involves interactions of amino acids in several portions of molecule.

“NT-gate” replaced with “Fast-Vj gate”.

nM sensitivity to Ca ions is an exaggeration; reports of effects below 100 nM are dubious, and most studies on hemichannels consider mM levels!

Replaced nM with: “low [Ca2+]i are effective in gating gap junction channels [13, 14, 18, 23, 28-40]”.

The sections dealing with modulation of Vj gating by chemical agents is somewhat novel and useful, but the section on the voltage sensitivity of the chemical gate and involvement of CaM are well covered territory by that author, including many reviews and even a full book.

Reviewer 2 Report

Comments and Suggestions for Authors

Thank you for considering and responding to  each of my suggestions for clarification or revision.  

Author Response

Thanks!

Round 3

Reviewer 1 Report

Comments and Suggestions for Authors

The authors has corrected most of the major concerns.  However, use of the term "likely" in the abstract is not consistent with views of others and should be replaced with "is hypothesized" or "may" or "is believed/proposed by this author".

There are a few instances where I believe that statements are misleading:

l201.  Yes, gating in response to a manipulation changing Ca was not abolished by deletion of the C terminus, but this mutation has been shown to attenuate pH sensitivity, indicating that a chemical gate resides within/relies on this domain.  The author is also no doubt aware that ODDD mutations in Cx43 that target the CaM binding region of the CL domain have been hypothesized to act through interference with kinase activity (Zheng, CHenavas et al, 2020), in contrast to the cork hypothesis.

l400.  This overstates the contribution of the slow gate to Vj channel closure in Cx45; it contributes, but fast Vj gating dominates.

Comments on the Quality of English Language

English is fine but copyeditor should check for redundant terms resulting from editing of sequential versions.

Author Response

The authors has corrected most of the major concerns.  However, use of the term "likely" in the abstract is not consistent with views of others and should be replaced with "is hypothesized" or "may" or "is believed/proposed by this author".

Replaced by: “We propose that the chemical gate is a calmodulin (CaM) lobe”.

There are a few instances where I believe that statements are misleading:

l201.  Yes, gating in response to a manipulation changing Ca was not abolished by deletion of the C terminus, but this mutation has been shown to attenuate pH sensitivity, indicating that a chemical gate resides within/relies on this domain.

Modified as follows: “Incidentally, also in Cx43 channels the truncation of most of the CT domain in mutant Cx43-M257 (Cx43K258stop) does not seem affect the chemical gating mechanism [28]. In contrast, data for a major role of CT in chemical gating by cytosolic acidification of Cx32 channels, via a ball-and-chain mechanism, have been reported [100]. Therefore, further work is needed to define in more detail the potential role of CT in chemical gating.

The author is also no doubt aware that ODDD mutations in Cx43 that target the CaM binding region of the CL domain have been hypothesized to act through interference with kinase activity (Zheng, CHenavas et al, 2020), in contrast to the cork hypothesis.

Added: “Significantly, a recent interesting study has demonstrated that mutations at Cx43’s CaM-binding site at CT, but not at CL2, in oculodentodigital dysplasia (ODDD), inhibit gap junction formation by limiting phosphorylation by Pyk2 and Src [120].” This study further confirms evidence for both a role of CaM in gap junction assembly [119] and evidence for CaM binding to CL2 and CT [42-44].

l400.  This overstates the contribution of the slow gate to Vj channel closure in Cx45; it contributes, but fast Vj gating dominates. With all due respect, this is not true. Please, read carefully the paper by Bukauskas et al, 2002, which I had cited (see below).

Evidence from Bukauskas et al (Bukauskas, F. F.; Angele, A. B.; Verselis, V. K.; Bennett, M. V., Coupling asymmetry of heterotypic connexin 45/ connexin 43-EGFP gap junctions: properties of fast and slow gating mechanisms. Proc. Natl. Acad. Sci. U. S. A 2002, 99, (10), 7113-7118.) clearly demonstrated that the slow gate has a major role, while the fast-Vj gate is only activated by large Vj gradients.

Citing here Bukauskas et al (2002): “Macroscopic and single-channel data indicate that the slow gates of Cx45 hemichannels dominate at small Vj values. Higher Vj values should activate the fast gates, which have faster kinetics”.

Also, from Bukauskas et al. (2002): “Because of the substantial fraction of channels closed by the slow gate of Cx45 at Vj = 0 mV, one might have predicted that in Cx45/Cx45 junctions, a still higher fraction of channels would be closed by this gate in response to Vj leaving little or no residual conductance (gmin).

Also, from Bukauskas et al. (2002): “. The two gates per hemichannel operate as if they are in series where the state of one gate affects the voltage across the other gate (“contingent gating”). In Cx45/Cx45 junctions at Vj = 0, a significant fraction of slow gates are closed, and the slow gate rather than the fast gate seems to be responsible for the strong Vj dependence.”

Also, from Bukauskas et al., (2002):At Vj = 0, the slow gate of many Cx45 hemichannels is closed in both homotypic Cx45/Cx45 and heterotypic Cx45/Cx43-EGFP junctions”. 

Comments on the Quality of English Language

English is fine but copyeditor should check for redundant terms resulting from editing of sequential versions.
